# Ocean-bottom and surface seismometers reveal continuous glacial tremor and slip

Evgeny A. Podolskiy [1,2 ✉], Yoshio Murai [3], Naoya Kanna [1,4] & Shin Sugiyama [1,2,5]

Shearing along subduction zones, laboratory experiments on analogue faults, and sliding along glacier beds are all associated with aseismic and co-seismic slip. In this study, an ocean-bottom seismometer is deployed near the terminus of a Greenlandic tidewater glacier, effectively insulating the signal from the extremely noisy surface seismic wavefield. Continuous, tide-modulated tremor related to ice speed is recorded at the bed of the glacier. When noise interference (for example, due to strong winds) is low, the tremor is also confirmed via analysis of seismic waveforms from surface stations. The signal resembles the tectonic tremor commonly observed during slow-earthquake events in subduction zones. We propose that the glacier sliding velocity can be retrieved from the observed seismic noise. Our approach may open new opportunities for monitoring calving-front processes in one of the most difficult-to-access cryospheric environments.

[1] Arctic Research Center, Hokkaido University, Sapporo, Japan. [2] Graduate School of Environmental Science, Hokkaido University, Sapporo, Japan. [3] Institute of Seismology and Volcanology, Faculty of Science, Hokkaido University, Sapporo, Japan. [4] Atmosphere and Ocean Research Institute, University of Tokyo, Kashiwa, Japan. [5] Institute of Low Temperature Science, Hokkaido University, Sapporo, Japan. ✉email: evgeniy.podolskiy@gmail.com

Recent discoveries of anomalously slow earthquakes and associated nonvolcanic tremor (commonly termed 'tectonic tremor') have revolutionized our understanding of the connection between slow and fast (regular) earthquakes, and their respective roles in fault-zone processes[1–5]. Analyses of weeks-long episodic tremor[1], which could resemble storm-generated noise[3], have now been surpassed by machine learning techniques that yield continuous, years-long seismic chatter along a subduction zone[6,7]. These findings raise the possibility of similar phenomena under polar glaciers[8,9], which slide at rates of up to four orders of magnitude higher than tectonic faults[10,11]. To date, only low-amplitude, discrete basal icequakes have been reported beneath the Greenland Ice Sheet (GrIS) and Antarctic ice streams[12–16]. Researchers have documented the teleseismically detectable $M_w$ 7 stick–slip of an Antarctic ice stream[17] and its associated tremor bursts[18,19]. However, such events have not been documented in Greenland, as $M_{S50}$ 4.6–5.1 events, which were initially termed 'glacial earthquakes'[20,21] because they were believed to emanate from the downhill sliding of the glacier trunk, have now been recognized as iceberg capsizing[22]. Further investigations of the seismic fingerprints of glacier sliding are motivated by (1) the possibility of estimating the frictional state of a dry laboratory fault from seismic signals[23], and (2) the results of field and cold-laboratory analyses, which support the use of basal seismicity to assess the coupling at the ice–bed interface[14,24–26], especially when the parameterization of glacier basal sliding remains the key uncertainty in predictions of ice mass loss and sea-level rise[10,11,27,28].

The slow-moving interiors of the polar ice sheets have been long recognized as regions with the lowest seismic noise levels on Earth, comparable to the most noise-free stations in the continental United States[29,30]. Consequently, GrIS was considered an ideal site to monitor the underground nuclear explosions conducted by the Soviet Union[31]. However, ice-breaking noise forced temporal termination of seismic monitoring in certain GrIS locations, such as coastal Thule[32]. Indeed, the Blue Trek seismic-noise survey across Northwest Greenland in August 1967 (Fig. 1a) revealed a trend of increasing short-period noise (<1 s) toward the coastline with increasing ice-flow speed[29,33]. Recent seismic observations near the calving front of Bowdoin Glacier (Kangerluarsuup Sermia in Greenlandic[34]), which is the subject of this study and could be seen as the endpoint of the Blue Trek survey (Fig. 1a and b), have revealed an extremely active seismic wavefield that is directly driven by crevasse opening due to tide-modulated strain-rate variance and indirectly driven by ice speed[35,36]. Therefore, if the basal slip is not aseismic, then it most likely produces weak seismic waves[3] that could simply be hidden in the noise. Before proceeding, we note that classification of the ambient seismic field as 'noise' persists as a contradiction between past efforts to suppress it and today's consideration of 'noise' as 'signal'[37]. We acknowledge this issue and use 'signal' and 'noise' interchangeably, depending on the context.

Other fundamental difficulties surrounding the labour-intensive and dangerous seismic monitoring at the calving front of a tidewater glacier that has an acoustically active supraglacial hydraulic system of streams and moulins[38] include melt-out (and therefore level loss), station drift due to ice flow (~40 m per month at Bowdoin Glacier), and the risk of instrument destruction due to crevassing and calving[8,35,36]. On-rock seismic stations are often installed on the lateral side of the glacier to avoid the latter issues[8]. However, while such a configuration is located away from the region of the fastest glacier flow and avoid the associated ice-generated noise, the seismic waves from the target area (the subglacial bed) must travel along a complex signal path and attenuation structure that consists of a multi-layered interface between ice, sediments, and bedrock, and, in the case of Bowdoin

Glacier, a lateral river that may serve as a potential fluvial seismic source (Fig. 1b). Finally, wind can increase the short-period noise levels of any surface station.

Here, we overcome the aforementioned challenges by the deployment of an ocean-bottom seismometer (OBS) near or under the calving front of Bowdoin Glacier (Fig. 1b–d). This glacier is one of ~242 marine-terminating glaciers in Greenland[10] where in situ data remains scarce due to extremely challenging access owing to the ice mélange, crevasses, kilometre-scale calving events, and calving-generated tsunami[39,40]. Due to the latter environmental constraints, weather, and complex logistics requiring coordinated operations on ice, rock, and ocean, our deployment was necessarily short. However, it was supplemented with seismic and ice-speed measurements at the ice surface (see the "Methods" section for details), and its findings were reproduced using the oldest seismo-geodesic records collected in 2015 at Bowdoin Glacier[35]. Therefore, the analysis results are of value due to a combination of both submarine and surface instruments and their unique proximity to the calving front. By 'eavesdropping' directly at the sliding base of this tidewater glacier, the OBS data reveal that the glacier continuously radiates seismic energy. A tremor-like signal is generated that lasts at least 2 weeks and resembles the Cascadia subduction zone tremor[6], but with strong diurnal and semi-diurnal periodicities. A comparison of the OBS signal with the horizontal displacement rate of the glacier demonstrates that the background noise may potentially serve as a proxy for the ice speed and ultimately the physical state of the ice–bed interface.

## Results and discussion

**Continuous seismic radiation from a glacier sliding**. Machine-learning studies have shown that the power of a seismic signal from both artificial and natural faults is the most important characteristic in extracting fault behaviour[6,23]. Furthermore, since we are assuming the overall similarity between a fault and the sliding glacier bed, we note that while the continuous tremor of the Cascadia subduction zone exists down to ~1 Hz, the signal in the 8–13 Hz band is nonetheless strong[6]. We first deconvolve the instrument response and compute the power spectral density–probability density functions (PSD–PDFs, after ref. [41]) to determine the statistical features of the continuous seismic data that best correlate with glacier dynamics while also keeping the signal processing as simple as possible. The representative velocity spectrum of the OBS data highlights that the horizontal components carry more power (see the "Methods" section for details), with an elevated energy level between approximately 0.6 and 14 Hz, and peaks at 0.7–2.0 and 7–14 Hz (Fig. 2a and b). Seafloor noise levels are high near the 1-s period due to a broad microseism peak[42]. This implies that the velocity spectrum could be relatively flat if it is not interrupted by the distinct peak near 1 s (Fig. 2a). However, above 5–10 Hz, the noise levels may be lower than those at the most noise-free continental borehole seismometers and therefore provide an advantage in the detection of local microearthquakes[42]. Note that, for such relatively high frequencies, the latter observation also implies lesser importance of any processes related to oceanographic pressure oscillations, which are of low frequency (see the "Methods" section for examples or ref. [43]). We investigate the seismic noise levels by computing the PSD for each 30-s segment of the OBS data and integrating the power in several frequency bands of interest that were chosen based on the PSD–PDFs analysis (0.1–0.6, 0.6–3.5, 3.5–7, and 7–14 Hz). We then present the temporal evolution of the normalized power in each band as density plots and extract the lowest noise level in 30-min-long sliding windows (Fig. 2c). This allows us to reveal the background diurnal signal above 0.6

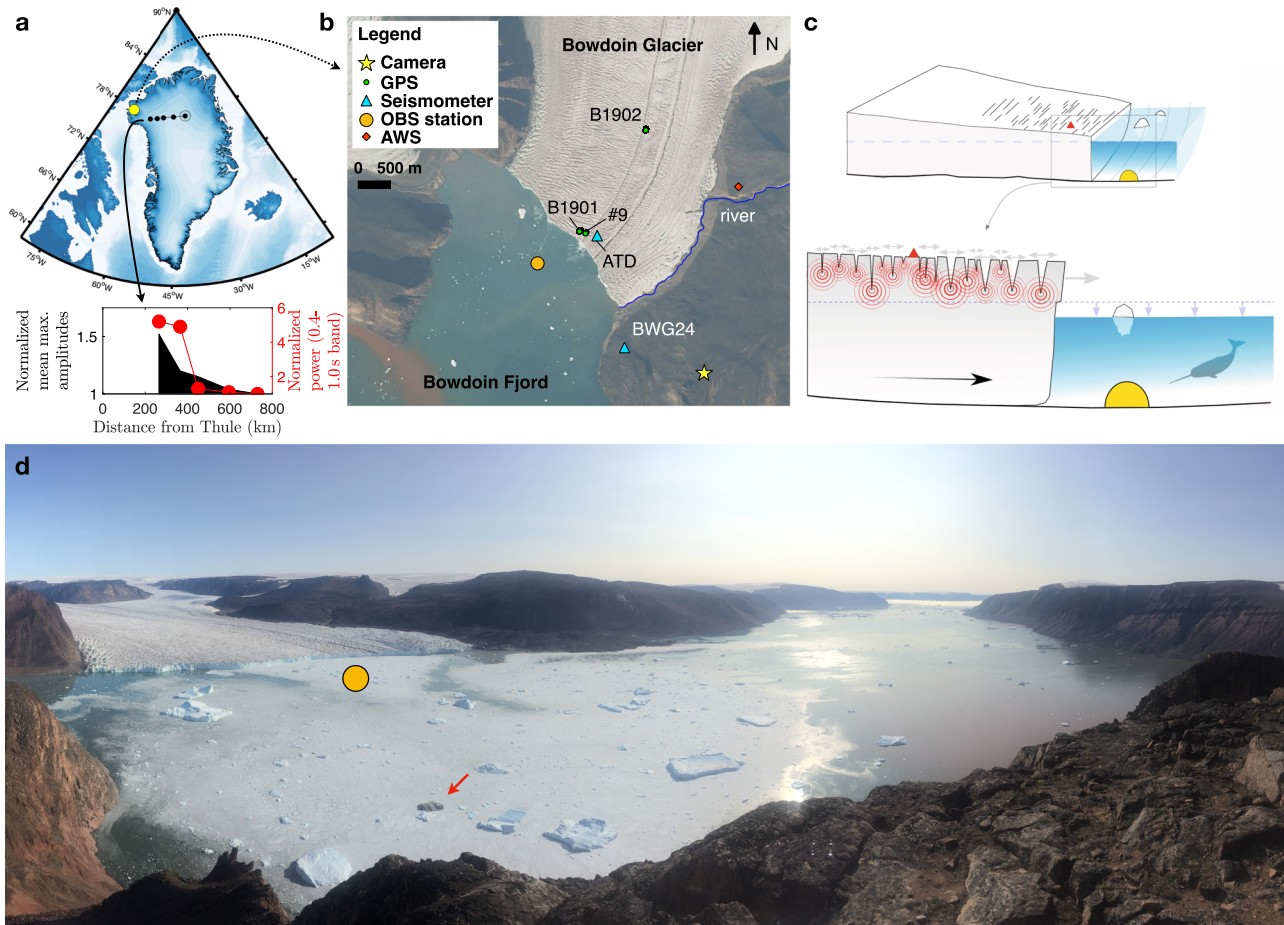

**Fig. 1 Details of the study site, 2019. a** Location of Bowdoin Glacier (yellow circle) relative to the 1967 Blue Trek seismic-noise survey (black circles), Greenland. The inset plot shows an increase in seismic noise from Inge Lehmann Station (open circle) to the coast; data are from ref. [29]. **b** Geophysical observational network on Bowdoin Glacier and its fjord (the background satellite image was acquired by Copernicus SENTINEL-2A on 27 July 2019; abbreviations correspond to the following: GPS Global Positioning System; OBS Ocean Bottom Seismometer; AWS Automatic Weather Station). **c** Schematic illustrating the key advantage of deploying an OBS directly under the calving front of a tidewater glacier to record subglacial and ocean seismo-acoustic signals, and minimise the impact of surface seismic sources (based on [35,36,48,58]). **d** Photograph of the fjord after a major calving event, with the approximate OBS drop point identified by the yellow circle (29 July 2019; credit: E. Podolskiy). An iceberg with a debris-laden base is marked with an arrow.

Hz that may be hidden between transient waveforms, which blur the statistics due to a high number of events (e.g., see Supplementary Fig. 1). The signal below 0.6 Hz appears to be dominated by ocean waves, with a stronger low-frequency signal observed during windy episodes (e.g., we observed strong winds on 23 July; Supplementary Figs. 2 and 3).

The broad-band seismic signal (3.5–14.0 Hz) correlates well ($R = 0.76$) with the global positioning system (GPS) displacement rates of the glacier that were measured ~880 m away (~150 m from the calving front; Fig. 3a and b) when we omit the potential microseism-polluted 0.6–3.5 Hz band. The obtained log-linear regression for predicting the ice speed, $v$, from the seismic signal, $T$ (i.e., $v = a\log_{10}(T) + b$) has an uncertainty range of $\pm 0.16$ m d$^{-1}$, which is equivalent to ~12% of the mean ice speed. We note that using the horizontal components does not lead to a significant improvement in the results.

We discuss below the 'basal-sliding-seismicity' hypothesis over competing hypotheses. Furthermore, in support of our arguments, we use surface stations on ice and rock located at different distances from the fastest moving part of the glacier in the same and previous years. Among the alternative sources of noise fluctuations, our key competing hypotheses are surface crevassing[35] and subglacial discharge[40,44,45]. Other continuous phenomena such as waves and processes in the ocean are difficult

to associate with the ice speed, whereas calving events are transient and relatively rare at Bowdoin Glacier[39]. Similarly, it is difficult to attribute the tide-modulated tremor, which is remarkably correlated with tide-modulated glacier velocities, to heterogeneous surface hydrology. For example, according to our direct observations, daily longitudinal and transverse traverses, and presence on Bowdoin Glacier, its relatively small streams of meltwater (flowing, remarkably, not only down-glacier, but also up-glacier[46]), moulins, and flow in crevasses are very local acoustic sources[38] that are unlikely to converge into a coherent, widely observed tremor.

There are two key differences between the observed continuous seismic signal and strain-rate variations that drive the intense microseismicity due to surface crevassing[35,36]. First, the correlation between the strain rate, $\dot{\varepsilon}$, and tremor is lower ($R = 0.45$), even if the peaks and troughs in the longitudinal extension that cause microseismic activity occur close to the peaks and troughs in ice speed (Fig. 3a). Second, the crevasse icequakes on Bowdoin Glacier are known to follow a strong 12-h tidal cycle of strain-rate oscillations[35,36], such that the strain-rate peaks are either similar in the morning and evening or stronger in the morning (Fig. 3a), whereas the ice speed is usually, but not always, higher in the evening due to delayed meltwater supply to the glacier bed[35,47] (Fig. 4). The latter pattern, which has more powerful evening

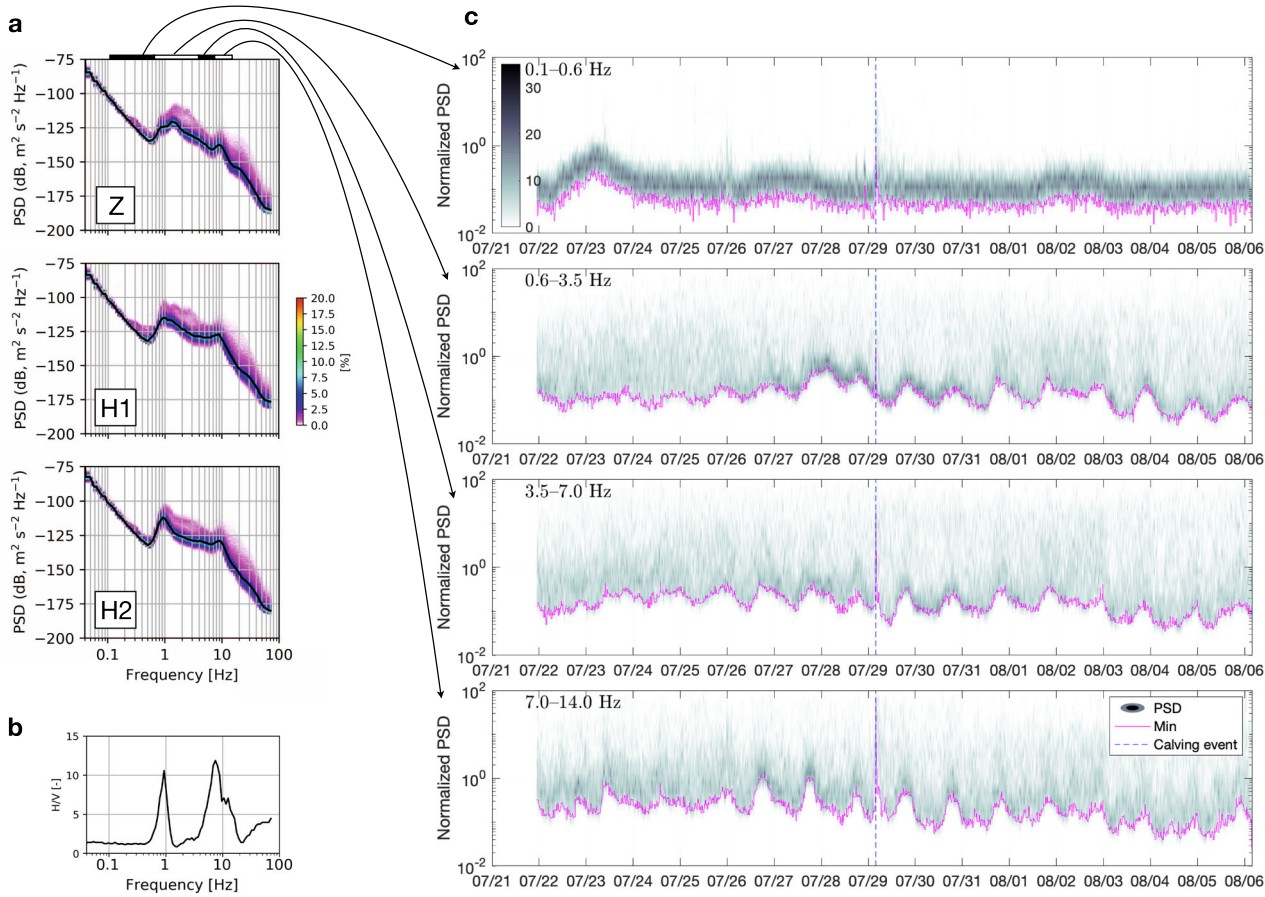

**Fig. 2 Power partition of the continuous ocean-bottom-seismometer (OBS) data for various frequency bands, 2019. a** Velocity spectra (power spectral density–probability density functions, PSD–PDFs) of the OBS three-component data. **b** Associated H/V spectral ratio (for accelerations). **c** Density plots of the normalized noise amplitude variation for all considered frequency bands. The magenta curve indicates the lowest noise level in each band. The dashed blue line indicates the timing of the major calving event on 29 July.

peaks, emerges in the tremor signal after 30 July (Fig. 3a) as the tidal amplitude recovers after a neap tide (Fig. 4), and is typical for the ice speed during this and previous years. Importantly, our sensitivity analysis (Supplementary Fig. 4 and Table 1) suggests that this diurnal pattern persists even at higher frequencies, after partial or complete exclusion of the possibly 'glacio-hydraulic' band (1–10 Hz[44]). Furthermore, such a double-hump (i.e., semi-diurnal) feature of tidal modulation is generally difficult to associate with subglacial discharge[44,45]. In this regard, a separate line of evidence suggesting that the subglacial discharge under Bowdoin Glacier is unlikely to dominate the observed broad-band tremor is provided by the first direct and continuous monitoring of a subglacial discharge plume near the centre of the calving front[40]. Specifically, even during such extreme hydraulic events as subglacial drainage of an ice-dammed lake, the associated seismic tremor remained primarily below ~1 Hz[40].

For the period of distinctly strong tidal modulation of the ice speed (i.e., with almost non-existent diurnal maxima; Fig. 3a), we found that the correlation of the low-frequency-band tremor (3.5–7.0 Hz; Fig. 2c) slightly improves with ice speed (0.84; not as expected for the diurnal melt cycle). However, the correlation decreases with strain-rate oscillations (0.27). The opposite trend is observed for the high-frequency band (8–20 Hz; Supplementary Fig. 4), with a decrease for velocity (0.67) and an increase for strain-rate oscillations (0.53). In short, the low correlation with strain rate and high correlation with the ice speed suggests that the tremor (including both 3.5–7.0 and 3.5–14.0 Hz bands) cannot be explained by surface crevassing. This further suggests

that high-frequency surface microseismicity is controlled by a distinct dynamic process that may saturate the seismic wavefield at on-ice stations (as shown below).

An exceptionally energetic, broadband anomaly was observed from 03:40 UTC on 29 July, when a 1-km-long iceberg calved off the centre of the glacier terminus and disintegrated 20 min later above the OBS station (Fig. 3c and d). The noise floor returned to its ambient noise level within several hours, despite the dynamic chaos of the ice mélange spanning the entire width of the fjord (Fig. 1d). This suggests that even the highest-magnitude and most prolonged seismic events of glacial origin, such as glacial earthquakes due to calving like the one discussed here and detectable more than 500 km away (Supplementary Fig. 5[20–22,48]), correspond to relatively short interference that masks the tremor signal. Fortunately, these calving events can be distinguished by their seismic signature having a particularly low characteristic frequency (Fig. 3c)[48] and may serve to motivate novel studies of the physics of calving, benefitting from near-field seafloor observations.

**Tremor signatures observed at surface stations.** We used this OBS tremor signal as a guide to performing a similar analysis of the continuous seismic data from the seismometers installed on the ice and rock on or near Bowdoin Glacier in July 2015 and July 2019 (Figs. 5–8). For overall consistency with the above analysis, the correlations were made with the 3.5–14.0 Hz band, unless specified otherwise (lower frequencies are shown in Supplementary Figs. 6–9). The 2015 on-rock station (CFH) exhibits a

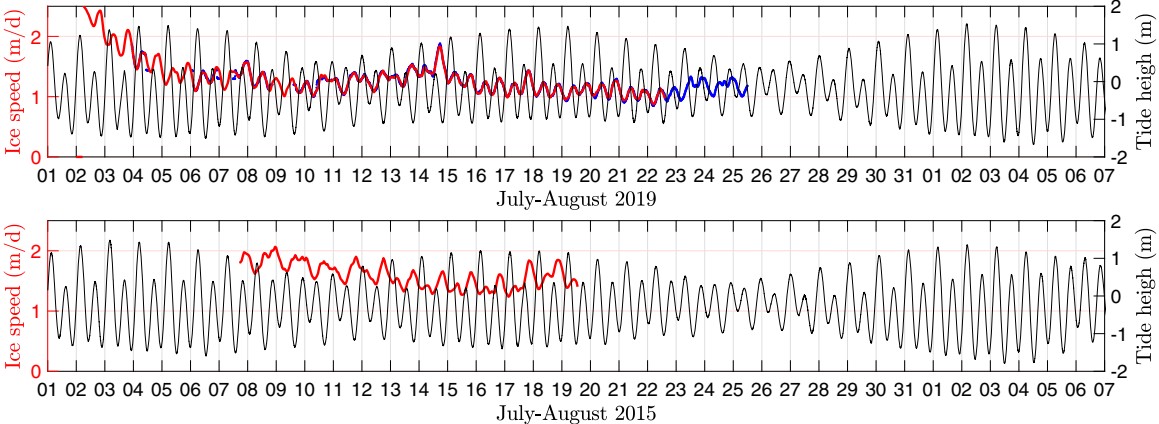

**Fig. 3 Covariance between tremor amplitude and ice speed, 2019. a** Seismic noise minimum amplitude (3.5–14.0 Hz power spectral density, PSD) versus glacier displacement rate at Global-Positioning-System (GPS) stations #9 and B1902, and the corresponding strain rate anomaly, $\dot{\epsilon}$ (i.e., deviation from the mean). The predicted ice speed for the period with missing GPS data is shown in green; the 95% confidence interval (in grey) yields a 0.16 m uncertainty in the estimated speed. **b** Scatter and density plots showing the correlation between the seismic signal and horizontal displacement rate at GPS #9 for overlapping data (R is the Pearson correlation coefficient). The red line is the regression fit. **c** Calving event as the cause of the main transient noise anomaly on 29 July (the signal is bandpass filtered at 1–10 Hz; the root-mean-square envelope with a 1-min moving window is shown in red; the times are in UTC). **d** Associated time-lapse images and their differences (i.e., direct subtraction of the greyscale intensity; the colour scale is in relative units, from 0 to ~100; the times are in UTC). The OBS drop location is marked by a triangle.

**Fig. 4 Tidal and ice-speed data.** Tide-gauge records of 2019 and 2015 from Pituffik/Thule versus the ice speed measured near the calving front of Bowdoin Glacier (GPS B1501 and GPS B1901/#9; the latter is shown in blue).

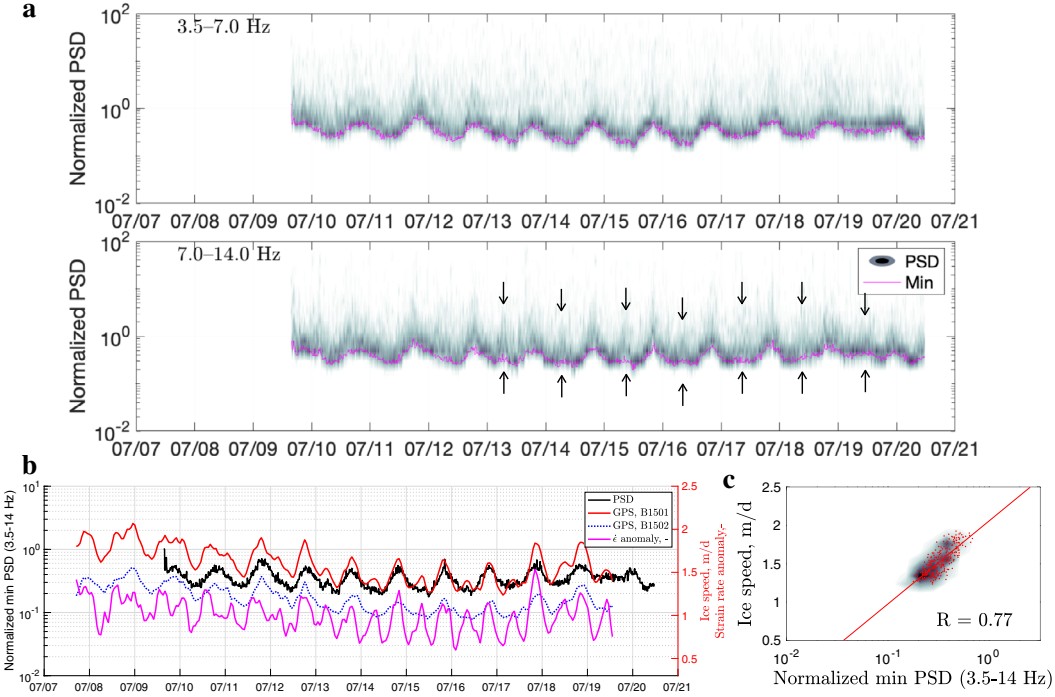

**Fig. 5 Analysis of the 2015 on-rock seismic station (CFH). a** Density plots of the normalized noise amplitude variations for 3.5–14.0 Hz frequency bands (for velocity power spectral density, PSD). Each magenta curve marks the lowest noise level in a given frequency band. Black arrows indicate examples of the morning peaks. The colour scale is the same as in Fig. 2c (0–35). **b** Minimum seismic tremor amplitude (3.5–14.0 Hz) versus glacier displacement rate at Global-Positioning-System (GPS) stations B1501 and B1502, with the corresponding strain rate anomaly, $\dot{\varepsilon}$ (i.e., deviation from the mean). **c** Scatter and density plots showing the correlation between the seismic signal and horizontal GPS displacement rate at B1501 for overlapping data ($R$ is the Pearson correlation coefficient). The red line is the regression fit.

remarkable overall correlation between ice speed and noise-floor variation (0.77; Fig. 5), while detailed visual inspection of the 7–14 Hz noise-amplitude variations (Fig. 5a) reveals tidal modulation, consistent with minor morning peaks in ice speed that are clearly unrelated to meltwater (e.g., 12–19 July 2015). The 2019 on-rock station (BWG24) exhibits an overall poor correlation (0.2; Fig. 6), but neither its noise-floor signal nor the 7–14 Hz noise-amplitude variations reflects the tidal cycle (except 7–9 July 2019; Fig. 6a). The noise-floor variation at BWG24 generally has a diurnal periodicity resembling that of the up-glacier speed (B1902). This poor correlation is likely due to strong winds exceeding 8 m s$^{-1}$ on 14 July (immediately before a speed-up), 20 July (during the calving of a 700 m × 50 m iceberg[49]), and 23 July (Supplementary Fig. 2). Further research is needed to clarify how the seismic sources between the station and the fastest section of the glacier (i.e., river, stagnant, and grounded ice) and/or increased buoyancy of the main terminus due to thinning[49] affect year-to-year differences. Nevertheless, the on-ice stations show a strong correlation between the tremor signal and the GPS displacement rate during both years (0.70 and 0.64, respectively; Figs. 7 and 8). Without the OBS data, it would be difficult to recognize this correlation at the 2015 and 2019 on-ice stations (ICC and ATD, respectively) because the strain rates are correlated almost equally well (as is ice speed) with the tremor (0.69 and 0.61), whereas the on-rock stations exhibited weaker correlations with $\dot{\varepsilon}$ (0.56 and 0.15). This demonstrates that intense near-surface fracturing is important on the ice, as shown by a comparison using a higher-frequency band (8–20 Hz; Supplementary Fig. 4), where ice speed exhibits a weaker correlation than strain rate at the on-ice stations during both years (ICC: 0.51 versus 0.61; ATD: 0.63 versus 0.66). However, the opposite trend is observed at the on-rock stations, with the velocity exhibiting a

stronger correlation than the strain rates (CFH: 0.81 versus 0.7; BWG24: 0.23 versus 0.16), which is likely due to attenuation of glacier surface microseismicity and the lack of coupling between glacier surface and the ice-free coastal area. In this regard, we note that the lateral ice margins of Bowdoin Glacier are elevated to ~20 m above the coast[46].

We also note that our consideration of higher frequencies (11–20 Hz), which do not overlap with an earlier proposed subglacial-discharge band (1–10 Hz[44]), yields similar patterns and correlation results (Supplementary Table 1). It is of particular interest that such a 'non-hydraulic' high-frequency band (>8–11 Hz) still exhibits a diurnal pattern with a maximum in the evening, when the ice speed is typically highest (see the OBS data between 26 July and 2 August 2019 or the CFH data between 11 and 18 July 2015 in Supplementary Fig. 4).

In contrast to the 2019 on-rock station (Fig. 6), the on-ice seismometer better resolved the speed-up event of 14 July due to less wind noise, while ice speed was poorly resolved during another windy episode (23 July; Fig. 8). Overall, we find the OBS tremor signal to be superior to the seismic signals from the surface stations due to its proximity to the tremor source and vertical distance from the noisy surface seismic wavefield. In a sense, our OBS corresponds to a borehole seismometer deployed without any labor-costly and dangerous drilling. Moreover, the widespread presence of the tremor and its correlation with ice speed at surface stations precludes any hypothetical OBS data contamination by water-property variations near the calving front, where subglacial water upwells to the surface[40]. This is consistent with the measured water temperature at the bottom of the fjord (see the "Methods" section) being relatively stable and corresponding to cold (−1.8 °C) Polar Water[48,50] during the period of our OBS observations.

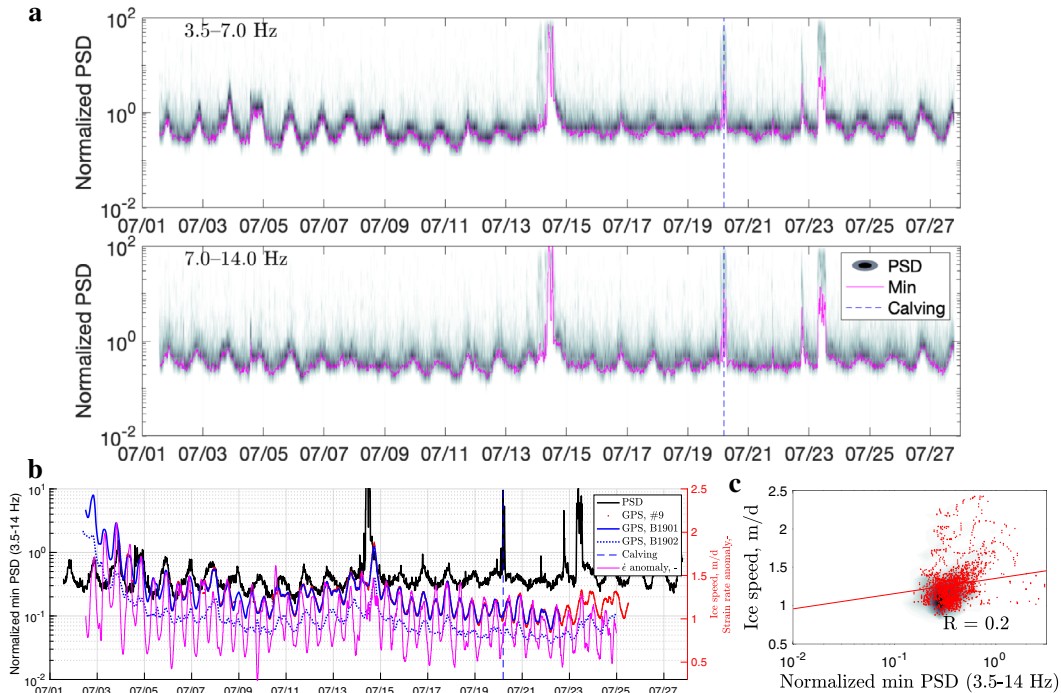

**Fig. 6 Analysis of data from the 2019 on-rock station (BWG24). a** Density plot of the normalized noise amplitude variations for 3.5–14.0 Hz frequency bands (for velocity power spectral density, PSD). Each magenta curve marks the lowest noise level in a given frequency band. The colour scale is the same as in Fig. 2c (0–35). The dashed blue line indicates the timing of a major calving event on 20 July. **b** Minimum seismic tremor amplitude (3.5–14.0 Hz) versus glacier displacement rate at Global-Positioning-System (GPS) stations B1901, B1902, and #9, with the corresponding strain rate anomaly, $\dot{\epsilon}$ (i.e., deviation from the mean). The dashed blue line indicates the timing of a major calving event. **c** Scatter and density plots showing the correlation between the seismic signal and horizontal GPS displacement rate at B1901 for overlapping data ($R$ is the Pearson correlation coefficient). The red line is the regression fit.

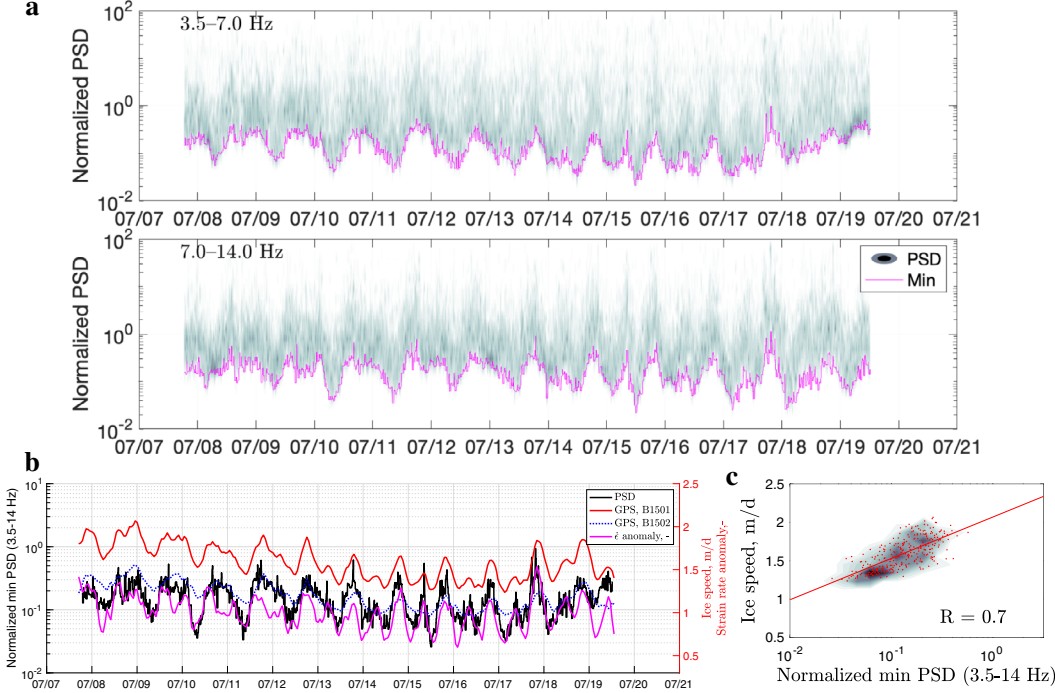

**Fig. 7 Analysis of data from the 2015 on-ice station (ICC). a** Density plots of the normalized noise amplitude variations for 3.5–14.0 Hz frequency bands (for velocity power spectral density, PSD). Each magenta curve marks the lowest noise level in a given band. The colour scale is the same as in Fig. 2c (0–35). **b** Minimum seismic tremor amplitude (3.5–14.0 Hz) versus glacier displacement rate at Global-Positioning-System (GPS) stations B1501 and B1502, with the corresponding strain rate anomaly, $\dot{\epsilon}$ (i.e., deviation from the mean). **c** Scatter and density plots showing the correlation between the seismic signal and horizontal GPS displacement rate at B1501 for overlapping data ($R$ is the Pearson correlation coefficient). The red line is the regression fit.

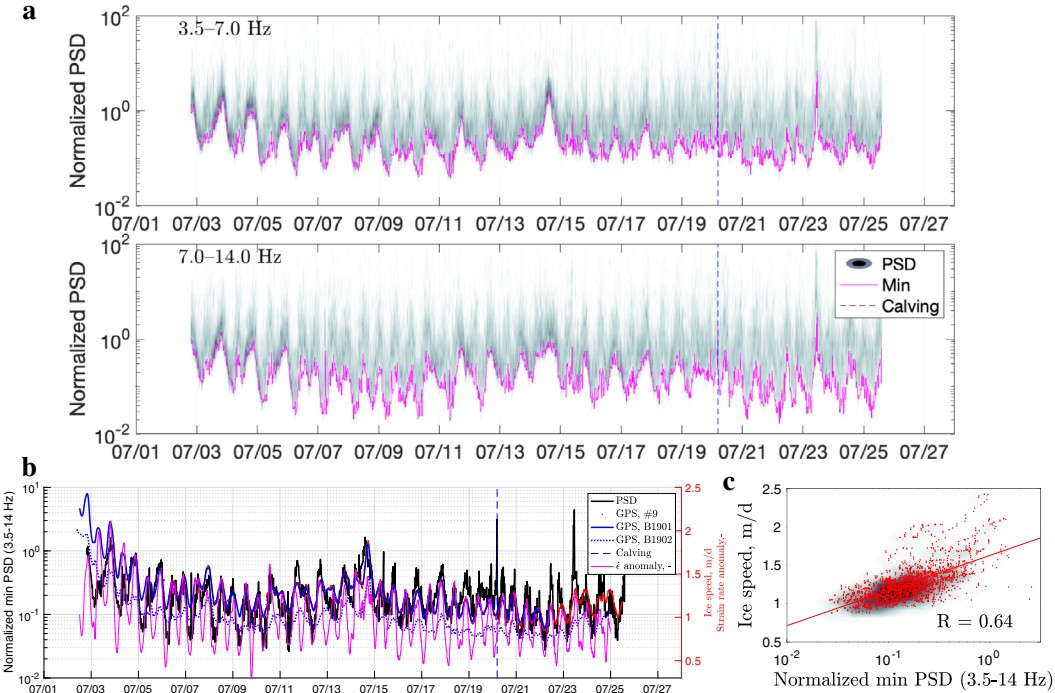

**Fig. 8 Analysis of data from the 2019 on-ice seismic station (ATD). a** Density plots of the normalized noise amplitude variations for 3.5–14.0 Hz frequency bands (for velocity power spectral density, PSD). Each magenta curve marks the lowest noise level in a given frequency band. The colour scale is the same as in Fig. 2c (0–35). The dashed blue line indicates the timing of a major calving event on 20 July. **b** Minimum seismic tremor amplitude (3.5–14.0 Hz) versus glacier displacement rate at Global-Positioning-System (GPS) stations B1901, B1902, and #9, with the corresponding strain rate anomaly, $\dot{\epsilon}$ (i.e., deviation from the mean). The dashed blue line indicates the timing of a major calving event. **c** Scatter and density plots showing the correlation between the seismic signal and horizontal GPS displacement rate at B1901 for overlapping data ($R$ is the Pearson correlation coefficient). The red line is the regression fit.

**Glacier slip as part of the slow-earthquake family.** Available data are limited but analysis of those from stations at different distances from the fastest moving section of Bowdoin Glacier consistently reveals the presence of the displacement-rate-correlated tremor within the frequency band suggested by a tectonic study[6]. Although this may be a coincidence, the band-limited noise is difficult to attribute solely to attenuation because our off-ice stations do not miss high-frequency seismicity (e.g., >8 Hz, Supplementary Fig. 4; or >11 Hz; Supplementary Table 1). The strong correlations with the GPS displacement rate and the close resemblance between the dominant frequencies of the Bowdoin Glacier tremor and tectonic tremor observed in Japan and Canada (1–10 and 8–13 Hz, respectively[1,3,6]) suggest similar physical processes, whereby the observed tremor is generated by slab sliding and friction under high pore-fluid pressure. We suggest that the elastic waves are generated at the ice–bed interface via either entrained-sediment friction[24] or protruding clasts, which are dragged by the sliding glacier and either plough/deform the water-saturated till or scratch the hard patches of the bed[11]. Rock-on-rock friction due to entrained sediments was recently proposed as the most likely cause of subglacial shear seismicity, as inferred from cold-laboratory experiments[24]. Photographs of calved icebergs (Fig. 1d) show that the basal ice is debris-laden, which is a requirement for the rate-weakening behaviour resulting from rock-on-rock friction and particle ploughing[24,25]. These results further suggest strong parallels with the commonly assumed slip zone fault model of elastic patches embedded into a velocity-strengthening, viscous background[51]. Only the GPS stations near the calving front show a pronounced tidal peak, as this peak is either missing or weak 1.8 km upglacier[35,47] (Fig. 6), and the southeastern part of the glacier is shallow and flows at half the

ice speed[49,52]; therefore, the southeastern part is unlikely to be the major radiator. The elastic patches should be located within the fastest-moving section of the glacier trunk, northwest of the central moraine. Such daily dislocation is equivalent to a cumulative $M_w$ 4.6 event[17], which is atypical of ordinary earthquakes in terms of duration but is characteristic of slow earthquakes[8,53].

Glacier tremor is stronger and more clearly correlated with the GPS data than subduction-zone tremor[6], presumably owing to the significantly higher displacement rates in glacier systems. However, the contrast in timescales used for comparison (days versus years) may partly explain this result (e.g., there may be a reduction in correlation due to noisy episodes, which are likely to occur in the longer term). Another difference we expect is that glaciers can flow at high velocities without a proportionate increase in seismic noise, as a result of either partial or full decoupling of the glacier from its bed at particularly low effective pressures. Specifically, recent cold-laboratory experiments and theoretical models exploring ice-on-till friction indicate that the rate-weakening friction is a necessary condition for basal seismicity[24]. Relatively low effective pressures at the sliding interface may correspond to reduced frictional weakening[24]. For example, such decoupling can occur when there is insufficient development of the subglacial system to remove excess water pressure lifting the ice, or when the western zone of the Bowdoin Glacier transitions from near-floating to floating in the future[47,49]. Such a major loss of effective pressure is difficult to imagine within a given subduction zone.

Finally, our findings are somewhat at odds with the interpretation of velocity-independent basal drag corresponding to basal-friction-independent glacier sliding[10]. However, this interpretation has been challenged in recent studies[11,28]. In fact,

as an alternative to traditional power-law basal rheologies relating slip velocity to basal stress, recent studies have proposed numerical ice-sheet modelling based on analogue experiments with glacial till[27]. Such modelling indicates the validity of using Coulomb friction and shows reduced basal shear stress near the grounding line[27]. Furthermore, the shear stress is independent of a high slip velocity at low effective pressure because shallow bed deformation is initiated at the failure strength of the bed by particles dragged by the flowing ice[11]. Therefore, our conclusion that seismic radiation is related to sliding rate does not contradict our current understanding of the underlying physical processes (i.e., velocity-dependent basal drag of particles), at least from a conceptual perspective.

In summary, we present a seismic acquisition method (i.e., OBS) to circumvent the noisiness of calving-front environments while simultaneously harvesting glacioseismic information from continuous seismic data. This work reflects the recent paradigm shift in seismology from discrete to continuous analysis, and demonstrates that inconvenient 'noise'[32] contains a wealth of glaciological information. Furthermore, our approach is valuable in difficult-to-access glacier-fjord settings. This study may motivate long-term interdisciplinary studies employing similar integrated multi-purpose ocean acoustic systems to reveal tremor location, migration, triggering, and termination via external factors[1], as well as the presence of slow-earthquake phenomena (i.e., continuous co-seismic slip) in other glacial settings. The use of multiple OBS stations, rather than just one as in this study, would be particularly helpful in locating the tremor, although, considering the cost of OBS instruments and the difficulty of approaching calving fronts, single-station submarine experiments may be more feasible. Such observations may increase the understanding of the coupling between the ice and its subglacial bed, especially as many marine-terminating glaciers continue to thin and approach floatation.

## Methods

**OBS system.** A pop-up-type OBS station with an acoustic release system (previously used in subduction-zone and other tectonic studies offshore of Japan) was deployed near the glacier calving front[48,54–57]. This setup consisted of a velocity seismometer, recorder, and lithium batteries that were sealed in a glass sphere under vacuum, which was protected by a plastic shell and surrounded by autonomous external instrumentation (mainly a Mitsuya anchor unit and Kaiyo Denshi STH-10B acoustic transponder). The sensor consisted of a gimballed three-component geophone with a 4.5-Hz natural frequency (L-28LBH; Katsujima). The seismic data were digitised and sampled at 128 Hz, with the internal clock information assigned using a Katsujima HDDR2 recorder with a 16-bit A/D converter. The clock drift and instrument drift from the drop point were both negligible due to the relatively short and shallow deployment[48].

The system also included a radio beacon (RF-700A1; Novatech), and a compact hydrophone integrated with a recorder (SoundTrap ST300 STD; Ocean Instruments) that sampled at 96 kHz and an internal thermometer (one sample per minute) that measured a mean water temperature of $-1.8 \pm 0.1\,°C$ (± shows the standard deviation). The high-frequency hydrophone was included for recording the soundscape of Bowdoin Fjord, which is seasonally visited by narwhals[58]. The temperature at the bottom of the fjord provides valuable information for both understanding the performance of the system and detecting deep intrusions of warm Atlantic water, the latter of which is important for subaqueous ice melt and has previously been observed in Bowdoin Fjord[50].

The OBS station was manually deployed in the centre of Bowdoin Fjord (243 m water depth) on 21 July 2019 from two small boats, ~640 m from the calving front (Fig. 1b). The station recorded data from 23:00 UTC on 21 July to 03:42 UTC on 6 August, when the anchor was released, with the OBS then recovered on 9 August.

Details of the study region are provided in historical[46] and our previous publications[35,36,47–49,58]. Here we note that the grounding line is expected to correspond closely to the calving front (Fig. 1b); according to our in situ geodesic observations, Bowdoin Glacier is nearly grounded, with no major floating ice tongue and with little vertical tidal admittance[35,47,49], which is large at floating glacier tongues[59]. Furthermore, due to difficult access and scarce data, the detailed geological composition of Bowdoin Fjord is poorly known. However, our seismic noise analysis (as detailed below) suggests a possible presence of soft sediments of up to a few metres thick (Fig. 2b), highlighting potential OBS use for mapping calving-front environments.

**Other observations.** GPS stations were installed close to the calving front (e.g., B1901 and GPS#9 at 120–150 m) and farther upglacier (B1902 at 2 km) by attaching the antennas to stakes that were drilled into the ice. B1901 and B1902 were dual-frequency GEM-1 receivers, whereas GPS#9 was a single-frequency Emlid Reach M+ receiver. The stations near the calving front are described in the previous studies[35,47,49]. The horizontal ice speed was computed with centimetre precision every 15 and 10 min, respectively, and the data were processed following a previous study[47]. GPS stations B1901 and #9 measured essentially the same ice speed ($R = 0.99$), due to their close proximity to each other (<100 m), but both are used here to maximize the observed time series (the 22–25 July data from GPS#9 are incorporated into the B1901 time series when necessary for the time-series comparisons). We also note that the surface GPS records closely correspond to basal sliding due to low internal ice deformation near the calving front as commonly assumed in models[10,27] and directly measured in boreholes at Bowdoin Glacier[60,61].

A Lennartz LE-3D/BH 1-s borehole seismometer was deployed in a shallow borehole (~3 m below the ice surface) within 280 m of each GPS station and connected to a DATA-CUBE³ datalogger (Omnirecs) that sampled at 400 Hz. A Lennartz LE-3Dlite MkIII 1-s seismometer was installed on the coast near Bowdoin Glacier, ~630 m south of the calving front and 1.9+ km from the GPS stations; the sensor was buried in a shallow pit and connected to a Centaurus datalogger that sampled at 500 Hz.

Time-lapse images of the calving front (7377 × 4935 px) were taken every hour using a NIKON D800 camera that was positioned on the east side of Bowdoin Fjord from 2 July until 29 July.

GPS station B1501 was deployed in July 2015 at the same site as B1901 in 2019. A Lennartz LE-3D 5-s seismometer was deployed in a shallow ice pit 70 m from GPS station B1501, which was in approximately the same area as the 2019 on-ice station (ATD). The station was connected to a Guralp CMG-DAS-S6 digitiser that sampled at 500 Hz. A Guralp CMG40T 30-s seismometer was deployed on the coast at the same site as BWG24 of 2019. The station (CFH) was connected to a CMG-DAS-S6 recorder that sampled at 100 Hz. Further details of the 2015 campaign are provided in previous studies[35,36]. An overview of considered seismic data is provided as spectrograms in Supplementary Fig. 3 to illustrate the nontrivial nature of the hidden signals described in this paper.

An automatic weather station (AWS) was installed to the east of the glacier and operated from 1 to 28 July 2019; the details of the AWS have been reported previously[47]. An AWS was operated at the same site in July 2015. The median wind speeds in 2015 and 2019 were similar (0.9 and 1.0 m s$^{-1}$). However, the 2015 campaign had remarkably calm weather conditions that were favourable for ambient noise measurements[35,36].

Tidal-height data were obtained every 5 min at ~125 km from Bowdoin Glacier, at the Pituffik/Thule tide-gauge station (76.5434°N, 68.8626°W) and are shown in Fig. 4. Our previous comparisons of tidal data collected near the calving front of Bowdoin Glacier with the Pituffik data indicate they are very similar in amplitude and phase (e.g., $r^2 = 0.98$), and, due to contamination by local calving-generated tsunami signals, the latter is more convenient to process[35,39]. Detailed analysis of the tidal role in the Bowdoin Glacier ice speed and strain rate is provided in our previous studies[35,36,47,49]. In brief, the tide-modulated ice speed varies greatly, usually with two daily peaks corresponding to falling or low tide (Fig. 4). Moreover, considering the different types of glacial response to tides[35,59], we have demonstrated that, without in situ geodetical records of ice deformation, sources of tide-modulated microseismicity remain poorly constrained[35]. This further highlights that such rare records provide the most needed information.

**Power spectral density.** We estimated the PSD–PDFs[41] using continuous seismic data that were parsed into 360-s-long segments with a 50% overlap. We first computed the PSD for each segment after removing the instrument response over the 0.005–100 Hz range. For H/V analysis, we also differentiated the signal. We then smoothed the PSDs in 1/2-octave averages at 1/8-octave intervals, and collected the corresponding powers in 0.5-dB bins, which served as the basis for computing the PDFs. We present the PSD–PDFs in decibels relative to velocities.

We visualized the temporal variation of the power for different frequency bands by computing the PSDs using a 30 s window for each trace, and then integrating the corresponding power for the different frequency bands as follows:

$$T_i = \int_{f_1}^{f_2} \mathrm{PSD}_i(f)\,\mathrm{d}f, \qquad (1)$$

where the index $i$ denotes the time window, and $f_1$ and $f_2$ are the lower and higher frequencies of each band (with a width $\Delta f = f_2 - f_1$). The corresponding power is assigned to the central absolute time of each segment, and normalized by the bandwidth ($\times \Delta f^{-1}$) and its average logarithmic levels ($\times 10^{-\overline{\log_{10}(T)}}$).

**H/V spectral ratio analysis.** We estimated the horizontal-to-vertical (H/V) spectral ratio in two steps following a previous study[62]. We first computed the quietest one percentile of the noise for all three seismic components (i.e., as the output of the PSD–PDFs procedure). We then divided the average of the horizontal power (two components) by the vertical component (one component). This

revealed two distinct peaks (0.9 and 7.4 Hz) where the horizontal motion was stronger (Fig. 2b).

The OBS horizontal components may be stronger than the vertical component due to shear mode resonance in soft sediments and noise. The following three types of OBS noise are usually the strongest: microseisms driven by the local wind wavefield at frequencies near 1 Hz[42], tilt noise due to seafloor currents and compliance noise due to pressure variations caused by ocean gravity waves[63]. We suggest that 0.9-Hz peak is due to microseisms (i.e., 1.8 s fjord waves), whereas the 7.4-Hz peak is probably due to the site response, as the periodicity of ocean gravity waves is too long for our analysis and particularly strong seafloor currents are not expected (furthermore, these frequencies carry a time-varying power that is correlated with the ice speed). The sediment thickness, $H$, can therefore be inferred from the dominant frequency, $f$, and shear velocity, $V_s$, such that $H = \frac{V_s}{4f}$ ref. [64]. An assumed velocity between 25 m s$^{-1}$ ref. [42] and 100 m s$^{-1}$ ref. [65] yields a thickness of between 0.8 and 3.4 m.

## Data availability

The SENTINEL-2A satellite imagery was downloaded from https://earthexplorer.usgs.gov/. The seismic and geodesic data are publicly available through the Arctic Data archive System website (https://ads.nipr.ac.jp/dataset/; A20200108-002, A20200108-003, A20200108-006). The 2019 GPS and wind-speed records, the time-lapse imagery, and tidal-height data are provided in the Supplementary Data (1–6). Tide data and GLISN seismic data are also publicly available through the Global Sea Level Observing System network (http://www.ioc-sealevelmonitoring.org/) and IRIS, respectively. Regional surface-wave-velocity range (Supplementary Fig. 5) is adopted from ref. [66].

## Code availability

The analysis was conducted and the plots were produced using Matlab R2018b (https://mathworks.com/products/matlab.html), and the Matplotlib, ObsPy, and Pyrocko Python libraries[67–69]. We used existing open-source Python toolboxes for handling seismic data (https://docs.obspy.org/ and https://pyrocko.org/). For image processing (Fig. 3d), we used a custom Matlab code provided in the Supplementary Data 6[70]. The site maps (Fig. 1a and b) were generated in open-source QGIS software, version 2.18.27 and in M_Map, a publicly available mapping package for Matlab (https://www.eoas.ubc.ca/~rich/map.html).

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

## Acknowledgements

We thank E. van Dongen for providing the ice-speed measurements from GPS station #9 and time-lapse images, as well as insightful discussions on calving. We thank Air Greenland helicopter pilots and F. Lindner, A. Bauder, J. Wassermann, R. Genco, G. Lombardi, and M. Ripepe for logistical and technical support in 2015 and 2019. We thank F. Walter, and G. Jouvet for project support in 2019. We also thank colleagues (I. Asaji, T. Ando, Y. Sakuragi, A. Mangeney, and O. Castelnau) and local guides (T. Oshima, K. Petersen, Q. Aladaq, S. Aladaq, and R. Daorana) for their assistance during fieldwork. The Ministry for Nature, Environment and Justice (Government of Greenland) granted permission (C-19-37) to conduct this fieldwork. This work and N.K. were supported by an Arctic Challenge for Sustainability research projects (ArCS and ArCS-II; JPMXD1300000000 and JPMXD1420318865, respectively), funded by the Ministry of Education, Culture, Sports, Science and Technology of Japan (MEXT), J-ARC Net, Grants-in-Aid for Scientific Research "KAKENHI" No. 18K18175, and the Second Earthquake and Volcano Hazards Observation and Research Programme (Earthquake and Volcano Hazard Reduction Research funded by MEXT).

## Author contributions

E.P. conceived the experiment, E.P. and Y.M. prepared the OBS set-up. E.P., N.K. and S.S. conducted the fieldwork in Greenland. E.P. performed the analysis, prepared the figures, and wrote the manuscript. E.P., Y.M., N.K. and S.S. discussed the results and contributed to the writing.

## Competing interests

The authors declare no competing interests.
