## [Peer Review File · Nature Communications]

REVIEWER COMMENTS

Reviewer #1 (Remarks to the Author):

In this paper, the authors deploy an ocean-bottom seismometer (OBS) at the front of a tidewater glacier in Greenland. Part of the deployment overlaps with GPS data collection on the glacier both near the glacier front and several km upstream. By looking at OBS spectral power through time, the authors find a daily fluctuation in minimum power in two relatively high frequency bands (3.5-7 Hz, 7-14 Hz). The fluctuation appears to be correlated with inferred sliding velocity from recorded GPS motion for the three days that the GPS and OBS datasets overlap. On the basis of this correlation, the authors suggest that variations in seismic noise in these "tremor" bands results from varying basal frictional sliding rates and accompanying variations in seismic wave radiation. They relate these glacial seismic signals to tremor and low-frequency earthquakes inferred to similarly occur during slow (sub-seismic) sliding on earthquake faults.

The correlation between the tremor signal and inferred glacier basal slip rate is quite good. However, as presented, the argument that tremor seen on OBS results from frictional basal slip is not entirely convincing for several reasons:

1. The GPS and OBS datasets only overlap for three days. This is problematic because the correlation determined from just three days of data is central to the main text, as currently written. (I do note that the tremor/motion relationship does exist over longer time periods in seismic data from nearby rock seismic sites shown in the supplement, but this data should be in the main text if it's a critical part of why we should believe the correlation).
2. Could the tremor signal actually be hydrologic? The seismic noise increase is largely diurnal. Why couldn't it be an increase in hydrologic noise on the glacier rather than an increase in seismic wave radiation due to faster basal slip? This is hard to pick apart, because hydrology affects sliding speed. Could the authors find a way to rule out hydrologic noise as the source of the increased noise floor? (Sources of noise could be, eg., surface runoff, flow in cracks and moulins in the ice thickness, subglacial flow, which have power in bands similar to the tremor band you have identified). One thought: a hydrologic signal driven by a daily surface melt cycle would be primarily diurnal. Glacier slip rate, on the other hand, probably depends on both daily surface melt input to the bed plus tidally-driven changes in slip speed with a slightly different periodicity. Components of the noise signal that correlate with falling tide (not quite diurnal) regardless of the diurnal signal (which could be meltwater) could be more convincingly related to changes in slip speed only.

To my knowledge, use of an ocean-bottom seismometer to study a tidewater glacier in detail has not been done before. If the authors can convincingly show that the tremor results from frictional basal slip, this would be an important demonstration of a new way to instrument glaciers that reveals important but previously unobservable processes.

The included statistical analysis seems appropriate.

Lesser comments:

- It would be helpful to more clearly outline all the possible sources of the noise fluctuations, and why it can't be each.
- The tides clearly play a role in the glacier speed and strain rate data, but are not included. Is there nearby tidal height data or model output available that could be included in the figures? The tides

clearly play a role here, and the observations are hard to interpret without knowing what the tides are doing.

- The seismic noise data in the supplement is very helpful for the argument that the diurnal seismic source observed by OBS could be basal slip. I think it would be helpful to find a way to include it in the main text. Because OBS data can be affected by water temperature and pressure in complicated ways, these effects might be expected to be extra unpredictable near a glacier front where subglacial water of a different temperature with variable suspended load is discharged at different rates through time. I could imagine OBS data being contaminated by subglacial discharge in unpredictable ways, possibly on a diurnal basis, so the fact that you can see the tremor / slip correlation on rock sites rules out all this contamination, and is actually very important to show the signal is real and widespread.

- The comparison with slow earthquake processes is fascinating, but somewhat hard to follow logically.

- If you are using an OBS specifically because attenuation should remove most high frequency noise from surface crevassing, is it surprising that you see band-limited noise, similar to tectonic tremor? High frequency noise from basal slip will be diminished by attenuation just as well as high frequency noise from surface crevassing.

Line comments:

45 - What does it mean to "access the coupling"?

128-148 - I think there is an implicit argument here that the lower tremor band (3.5-7Hz) can't be from crevassing because it does not correlate with strain rate (while the upper tremor band does a bit more), but I'm not sure this is explicitly said. It should be, since this is an important part of the argument for why the diurnal 3-7.5 Hz tremor signal is basal sliding and not just crevassing.

154-157 - "Cryoseismic events" is really vague, and could include all kinds of signals that I don't think you're talking about here. Can you be more specific?

157 - What specifically about the seismic signature in the data presented?

165, 177 - What tremor band is the correlation for?

166 - What does not show the tidal cycle? The noise floor? In what band?

173 -176 - Aren't these opposite statements?

174 - Good correlation of what with what?

175 - Which signal? the diurnal variation in the noise floor?

176 - correlated equally well as what?

161-191 - This whole paragraph needs more specific language. It's not always clear which signals correlate with each other because of vague language like "the signal" rather than "the diurnal variation in noise floor" (or however you wish to say it).

179 - Where is the higher frequency band comparison shown? This paragraph would really benefit from having a visual summary of the supplement data in the main text. The points made are important but cannot be seen in the main text figures.

188 - This sentence says that the windy episode is poorly resolved, but I think you're trying to say that the seismic signal associated with the speedup is poorly resolved because of the wind. Clarify.

195 - What are the tremor bands in Japan and Canada?

206 - Remove word "that" ...?

214-216 - I don't understand why Mw4.6 event is atypical of ordinary earthquakes. Is it the daily duration that's atypical? Clarify.

217 - This is a confusing sentence - how can tremor be correlated with seismic data?

220-234 - I'm having trouble following the logic in this section

230-232 - Again, confusing. ...particles that drag at some low transition velocity?

236 - Name the "seismic acquisition method" (OBS?)

245 - What slow earthquake phenomenon are you referring to that you might see in other glacial settings?

Reviewer #2 (Remarks to the Author):

In their manuscript, the authors discuss data from a novel ocean-bottom seismometer (OBS) deployed very near the calving front of Bowdoin Glacier in Greenland. The OBS tremor data (~3.5-14 Hz) show remarkable correlation with glacier surface velocities, much higher than correlation with strain rate, suggesting that the tremor is directly related to slip speed rather than strain rate. The findings are very exciting especially in their constraint on subglacial physical processes, and demonstrate the usefulness of observing seismic tremor with near-glacier bed seismic deployments. After very minor revisions, the manuscript will be a very interesting contribution that will be of wide interest to the community. I further note that the analysis is well described, there is sufficient description to reproduce all of their figures, and there is truly no comparable study in the existing literature.

L48 The article "Marine ice sheet profiles and stability under Coulomb basal conditions" by Tsai, Stewart and Thompson (J Glaciol., 2015) would be very appropriate to cite here as a work on the same topic that predates the citations provided.

L107-109 Unclear how this statement can be substantiated or what the implication is. Please clarify.

Fig3 Why is GPS limited to such a short time? Is the end time when it calved and was lost? I don't question there was a good reason, but it seems strange to not comment on.

L146-148 I did not understand the logic of how this statement is related to the previous sentence. Clarifying would be appreciated.

L196 "indicates" is too strong and unsubstantiated. "suggests" would be fine. It is certainly possible that the phenomena are similar, but if the authors want to use "indicates" they need to demonstrate that tremor is generated by friction and high pore pressures, neither of which they appear to have direct evidence for. Perhaps this sounds like nitpicking to the authors, but it is my opinion that their exceptional data need not and should not be overinterpreted. Related to this, the frequency content may be more related to the distance than any similarities in the physics.

L228-232 Again, the 2015 J Glaciol. article cited above is very relevant and predates the citations provided. I encourage the authors to include this paper in their discussion.

-Victor Tsai

Reviewer #3 (Remarks to the Author):

This manuscript documented possible tremor like signals associated with glacier movements in Greenland as recorded by an OBS. There appears to be a very clear correlations between the tremor signal and ice speed. The authors “proposed that the degree of glacier sliding can be inverted from the observed seismic noise”, and their approach may provide a new way to monitor calving-front processes.

Overall I found this manuscript to be written well, and the results are interesting and exciting to justify a publication. However, I felt that the current version did not clearly convince me that the observed signals are similar to (or different from) the non-volcanic tectonic tremor observed at subduction zone and other tectonic settings. I believe that most of them can be accounted by replotting the observations (and rewriting), although the lack of additional OBS recordings is an issue that can only be addressed by future deployments. Below are my specific comments:

1. Due to its non-emergent features, tremor is relatively hard to observe, and typically multiple-station recordings are needed to demonstrate moveout and locate the source, before making further interpretations. However, in this study, only one OBS station is used. Hence, there is no way to confirm the observed seismic signals, and locate the seismic source, making the results less convincing. Since there are several nearby stations on land, perhaps the authors can at least try to plot a zoom-in waveforms (say within 1-2 hrs of a particular day) to demonstrate whether there are (or a lack of) some coherent seismic waves at these stations. In addition, the authors should also mention about the potential issue with single-station observations in the discussion part of the manuscript.
2. I would like to see a full spectrogram plot (i.e., frequency versus time) for the entire time period of the OBS recording (and perhaps several on-rock stations) to show readers the full information of what were recorded by these seismic stations (rather than only showing the PSD versus time for selected frequency range). A full spectrogram plot would be more informative and hopefully would show either similar or potentially more information than currently shown in Figure 2. This can be easily done using Matlab’s `specgram` function or the continuous wavelet transform (`ctw`) command.
3. Additional information is needed for readers to better understand the experiment setup environment and details of the study region. For example, do we know where the grounding line is, and how far it is from the OBS station? In addition, do we know the rock type and sediment thickness at the ocean bottom?
4. The authors mentioned in the abstract that “the tremor was also confirmed via an analysis of the seismic waveforms from surface stations”. If I remembered correctly, the authors stated that the tremors were hard to observe from surface stations, and did not show any correlations with the GPS movements. Please clarify. If the tremors can be observed by both the surface stations and OBS, then some plots/analysis as suggested in comment 1 should be performed. Finally, the authors stated that “The signal resembles the nonvolcanic tremor ...”. I wonder if they can make some plots to show some representative time series recorded by this station and those observed at subduction zones. In addition, in source seismology communities, we typically use “tectonic tremor”, rather than “nonvolcanic tremor” these days (<https://pubs.er.usgs.gov/publication/70192475>).
5. In the end of the abstract, the authors stated that “the degree of glacial sliding can be inverted from the observed seismic noise”. I am not sure what they meant by “the degree of glacial sliding”. Indeed, we saw a positive correlation between the surface GPS recording and the tremor-like signals. But surface GPS movement does not exactly correspond to the degree of glacial sliding, right? Some clarification is needed.

6. The authors used the term “signal” and “noise” interchangeably in the abstract and main text. It is clear that noise can be turned into signals. But I think that some simple definition at the beginning of the main text could be useful and will not confuse the readers.

7. Line 217, the authors wrote “Glacier tremor is stronger and more clearly correlated with the seismic data than subduction zone tremor [4]”. I am not sure what did they refer to as “seismic data”. Perhaps rather than “seismic data”, the authors meant “GPS data”?

8. The following references on the observations of seismic tremor associated with glacial movements, and reviews of global observations of fast and slow earthquakes are worth citing:

9. Winberry J. P., S. Anandakrishnan, D. A. Wiens, and

10. R. B. Alley (2013), Nucleation and seismic tremor associated with

11. the glacial earthquakes of Whillans Ice Stream, Antarctica, *Geophys.*

12. *Res. Lett.*, 40, 312–315, doi:10.1002/grl.50130.

Winberry J. P., S. Anandakrishnan, D. A. Wiens, and R. B. Alley (2013), Nucleation and seismic tremor associated with the glacial earthquakes of Whillans Ice Stream, Antarctica, *Geophys. Res. Lett.*, 40, 312–315, doi:10.1002/grl.50130.

Peng, Z. and J. Gomberg (2010), An integrative perspective of coupled seismic and aseismic slow slip phenomena, *Nature Geosci.*, 3, 599–607, doi: 10.1038/ngeo940.

Submitted: 16 November 2020, Invitation to revise: 19 January 2021

We thank Dr. Victor Tsai and two anonymous reviewers for evaluating our submission and encouraging us to improve the manuscript. We appreciate their thoughtful and helpful comments. The revised version of our manuscript is enclosed, and our detailed point-by-point responses are given below (shown in bold; modifications are highlighted in the manuscript). We believe these modifications and the 21 additional display units (including subplots and a table) have strengthened the paper.

Yours sincerely,
Evgeny Podolskiy and co-authors

REVIEWER COMMENTS:

Reviewer #1 (Remarks to the Author):

In this paper, the authors deploy an ocean-bottom seismometer (OBS) at the front of a tidewater glacier in Greenland. Part of the deployment overlaps with GPS data collection on the glacier both near the glacier front and several km upstream. By looking at OBS spectral power through time, the authors find a daily fluctuation in minimum power in two relatively high frequency bands (3.5-7 Hz, 7-14 Hz). The fluctuation appears to be correlated with inferred sliding velocity from recorded GPS motion for the three days that the GPS and OBS datasets overlap. On the basis of this correlation, the authors suggest that variations in seismic noise in these "tremor" bands results from varying basal frictional sliding rates and accompanying variations in seismic wave radiation. They relate these glacial seismic signals to tremor and low-frequency earthquakes inferred to similarly occur during slow (sub-seismic) sliding on earthquake faults.

The correlation between the tremor signal and inferred glacier basal slip rate is quite good. However, as presented, the argument that tremor seen on OBS results from frictional basal slip is not entirely convincing for several reasons:

1. The GPS and OBS datasets only overlap for three days. This is problematic because the correlation determined from just three days of data is central to the main text, as currently written. (I do note that the tremor/motion relationship does exist over longer time periods in seismic data from nearby rock seismic sites shown in the supplement, but this data should be in the main text if it's a critical part of why we should believe the correlation).

We agree that our confirmation of the tremor at other stations, especially in different years, is critical evidence that independently validates our findings. As advised, we have moved all of this visual data from the Supplementary Information to the main text, which has a specially dedicated section titled "Tremor signatures observed at surface stations". Furthermore, the surface stations are now mentioned not only in the Methods and Introduction, but also in the title to highlight the comprehensive nature of our work.

We understand that the surface stations make the findings more convincing, but we do not consider that the total length of our high-frequency observations should be used as a measure of their value. Records near the terminus are challenging due to calving of instruments into the fjord with kilometer-scale icebergs, ~30 m tsunamis generated by icebergs, and ice mélange blocking access to the fjord. To share our observers' views, we have further highlighted the unique nature of our simultaneous records. For example, we are not aware of any other glacier for which underwater/on-ice measurements have been taken within 640/200 m of the calving front.

2. Could the tremor signal actually be hydrologic? The seismic noise increase is largely diurnal. Why couldn't it be an increase in hydrologic noise on the glacier rather than an increase in seismic wave radiation due to faster basal slip? This is hard to pick apart, because hydrology affects sliding speed. Could the authors find a way to rule out hydrologic noise as the source of the increased noise floor? (Sources of noise could be, eg., surface runoff, flow in cracks and moulines in the ice thickness, subglacial flow, which have power in bands similar to the tremor band you have identified). One thought: a hydrologic signal driven by a daily surface melt cycle would be primarily diurnal. Glacier slip rate, on the other hand, probably depends on both daily surface melt input to the bed plus tidally-driven changes in slip speed with a slightly different periodicity. Components of the noise signal that correlate with falling tide (not quite diurnal) regardless of the diurnal signal (which could be meltwater) could be more convincingly related to changes in slip speed only.

Such an alternative interpretation, implying that the ice speed is remarkably correlated with subglacial discharge, would be an interesting achievement in itself. In this regard, we have included more discussion and analysis to demonstrate a lack of support for such a hydraulic hypothesis, based on the following reasoning:

(1) We report a tide-modulated tremor correlated with tide-modulated ice speed (Figs. 2c, 3a), although subglacial discharge is not a semi-diurnal process. For the data constrained by GPS, even within the so-called 'glacio-hydraulic' band (3.5-7 Hz), we observed a tide-modulated, semi-diurnal signal.

(2) By analyzing high frequencies (above the glacio-hydraulic band of 1-10 Hz), we show that semi-diurnal and diurnal features remain (Suppl. Fig. 5 & Table 1).

(3) Even during such extreme hydraulic events as subglacial drainage of a lake at Bowdoin Glacier we detected (in our specially designed experiment for continuous monitoring of a subglacial discharge plume at the calving front) a local tremor primarily below (not above) 1 Hz [kindly refer to Fig. 3 in the attached manuscript; *Podolskiy et al.*, under consideration].

(4) In our understanding, there is no previous observational or theoretical evidence that turbulent subglacial discharge could control ice speed. We acknowledge that subglacial water pressure rather than water flow is important for short-term ice speed variations. Large amounts of subglacial water may produce high pressures, but after a drainage conduit develops the water drains at low pressure. With well-developed subglacial plumes draining the water, we do not expect particularly high pressures other than perhaps during the first days of July 2019 (Suppl. Fig. 4).

We have also discussed the unlikelihood of other hydraulic near-surface sources (addressing another comment below). Furthermore, regarding the final sentences provided by the Reviewer, throughout the manuscript we have pointed reader attention to places with a correlation between components of the noise signal and slip, which were outside the main evening peaks (station CFH is outstanding in this regard). Finally, we have provided the tidal data (Suppl. Fig. 4). We hope this better addresses the question.

To my knowledge, use of an ocean-bottom seismometer to study a tidewater glacier in detail has not been done before. If the authors can convincingly show that the tremor results from frictional basal slip, this would be an important demonstration of a new way to instrument glaciers that reveals important but previously unobservable processes.

We believe our revisions have helped in this regard and that the manuscript has sufficient background to show that this pioneering work is not limited to basal processes. Rather it has broad potential to extend into other areas of cryoseismology to study calving and other types of glacial seismicity and for site characterization (including means of keeping multi-purpose instruments safe in extreme polar environments).

The included statistical analysis seems appropriate.

Lesser comments:

- It would be helpful to more clearly outline all the possible sources of the noise fluctuations, and why it can't be each.

Thank you for this advice. We have outlined the possible sources and highlighted why each is unlikely throughout the text, based on the diversity of presented features.

- The tides clearly play a role in the glacier speed and strain rate data, but are not included. Is there nearby tidal height data or model output available that could be included in the figures? The tides clearly play a role here, and the observations are hard to interpret without knowing what the tides are doing.

We included such openly available tidal data in supplementary figures (Suppl. Fig. 4) and data, with the necessary explanations in the Methods of the validity of these data and our studies in this regard (plus a link to the source in Data Availability). As highlighted in our previous paper, interpretations of tide-modulated glacial dynamics are 'blind' without *in situ* GPS observations of corresponding glacier deformation. The latter are necessary primary information. For example, due to the different responses of glaciers to tides, tidal variations alone had left the pre-dating interpretations of tide-modulated seismicity poorly constrained [*Podolskiy et al., GRL, 2016*].

- The seismic noise data in the supplement is very helpful for the argument that the diurnal seismic source observed by OBS could be basal slip. I think it would be helpful to find a way to include it in the main text. Because OBS data can be affected by water temperature and pressure in complicated ways, these effects might be expected to be extra unpredictable near a glacier front where subglacial water of a different temperature with variable suspended load is discharged at different rates through time. I could imagine OBS data being contaminated by subglacial discharge in unpredictable ways, possibly on a diurnal basis, so the fact that you can see the tremor / slip

correlation on rock sites rules out all this contamination, and is actually very important to show the signal is real and widespread.

We agree that our confirmation of similar signals at surface stations (particularly in different years) is a powerful illustration worth including in the main text. Therefore, we have moved surface-station plots from the Supplementary Information to the main text (the total number of main-text figures is now seven). Regarding the possible effects of water on OBS, in the Methods and in the main text we have clarified that during the period of our observations, the measured water temperature (-1.8°C) was relatively stable ($\pm 0.1^{\circ}\text{C}$), clearly corresponding not to subglacial discharge but to the cold Polar Water mass. We briefly included the corresponding discussion in the section about the surface stations, and mentioned that oceanographic pressure oscillations are known as low-frequency processes. Regarding the discharged suspended load (e.g., ~ 0.1 gram per liter), we feel it is premature to raise this in the manuscript because the measured turbidity of water in Bowdoin Fjord drops significantly below 50 m depth, while the suspended sediment load drops with distance from the plume [Kanna et al., JGR, 2018].

-The comparison with slow earthquake processes is fascinating, but somewhat hard to follow logically.

We are delighted to hear it is fascinating. We hope our clarifications and increased detail on recent glaciological studies in response to this and other minor remarks regarding the same section make it easier to follow.

- If you are using an OBS specifically because attenuation should remove most high frequency noise from surface crevassing, is it surprising that you see band-limited noise, similar to tectonic tremor? High frequency noise from basal slip will be diminished by attenuation just as well as high frequency noise from surface crevassing.

We have outlined in the Introduction that we propose this pioneering OBS approach for multiple reasons; for example, to avoid the physical destruction of a sensor, its melt out and level loss, and noise due to wind and other near-surface processes. We acknowledge the attenuation issue in the main text. From borehole seismology in tectonic and glacial settings, it is generally known that deep location of a sensor is helpful in avoiding surface-related processes. OBS deployment is equivalent to borehole seismology in this sense. In our case, the intended geometry of the OBS set-up corresponds to a shorter distance, L , to the glacial base than to the glacial surface ($L + h$, where h is the ice thickness). Furthermore, we mentioned that the tremor-band is recognized after broad-band analysis of stations located at different distances (0 m, ~ 880 m, and \sim almost 2 km) from the fastest moving section of the glacier which was sliding in correlation with the noise.

45 - What does it mean to "access the coupling"?

We have corrected this to "assess the coupling at the ice-bed interface".

128-148 - I think there is an implicit argument here that the lower tremor band (3.5-7Hz) can't be from crevassing because it does not correlate with strain rate (while the upper tremor band does a bit more), but I'm not sure this is explicitly said. It should be, since this is an important part of the argument for why the diurnal 3-7.5 Hz tremor signal is basal sliding and not just crevassing.

We have added a summary phrase to this text.

154-157 - "Cryoseismic events" is really vague, and could include all kinds of signals that I don't think you're talking about here. Can you be more specific?

We have specified events as being seismic events of glacial origin, particularly, calving.

157 - What specifically about the seismic signature in the data presented?

This part was referring to the literature; for the presented data, we explain that it is low-frequency content.

165, 177 - What tremor band is the correlation for?

In the beginning of the corresponding paragraph, we have specified that we refer to the previously used broad band using a "similar analysis".

166 - What does not show the tidal cycle? The noise floor? In what band?

Yes, we have specified this. After modification regarding the previous comment, the band should now be clear from the context.

173 -176 - Aren't these opposite statements?

Thank you for highlighting this ambiguity. We have clarified that our intended meaning was referred to the difficulty of recognizing the signal by using on-ice stations alone.

174 - Good correlation of what with what?

We have clarified that we mean between the seismic data and the GPS displacement rate.

175 - Which signal? the diurnal variation in the noise floor?

We have clarified that we refer to the tremor correlated with the GPS displacement rate.

176 - correlated equally well as what?

We have specified that it was the ice speed.

161-191 - This whole paragraph needs more specific language. It's not always clear which signals correlate with each other because of vague language like "the signal" rather than "the diurnal variation in noise floor" (or however you wish to say it).

We have modified this paragraph and hope that the language is now less ambiguous.

179 - Where is the higher frequency band comparison shown? This paragraph would really benefit from having a visual summary of the supplement data in the main text. The points made are important but cannot be seen in the main text figures.

Initially, our intention was to highlight points by presenting the correlation as a standard regression metric, rather than by adding extra supplementary figures. As major rearrangements of the latter have been made, we have added 10 extra subplots for the higher-frequency comparison. As suggested in another comment by the Reviewer, the original supplementary illustrations were integrated directly into the main text to make the corresponding paragraph easier to follow.

188 - This sentence says that the windy episode is poorly resolved, but I think you're trying to say that the seismic signal associated with the speedup is poorly resolved because of the wind. Clarify.

Yes, thank you for highlighting this. We have clarified that we refer to the ice speed.

195 - What are the tremor bands in Japan and Canada?

Specified.

206 - Remove word "that"...?

Removed.

214-216 - I don't understand why Mw4.6 event is atypical of ordinary earthquakes. Is it the daily duration that's atypical? Clarify.

Yes, we have clarified that the duration is atypical.

217 - This is a confusing sentence - how can tremor be correlated with seismic data?

Corrected to “GPS data”.

220-234 - I'm having trouble following the logic in this section

To make it easier to follow, we have provided more details, broken this text into two paragraphs, and accommodated other minor suggestions including incorporation of additional references.

230-232 - Again, confusing. ...particles that drag at some low transition velocity?

We removed details from the cited paper proposing this and simplified it to “particles dragged by the ice flow”.

236 - Name the "seismic acquisition method" (OBS?)

Yes; specified.

245 - What slow earthquake phenomenon are you referring to that you might see in other glacial settings?

We have specified that we mean a continuous co-seismic slip.

Once again, we thank the Reviewer for help to improve our manuscript.

Reviewer #2 (Remarks to the Author):

In their manuscript, the authors discuss data from a novel ocean-bottom seismometer (OBS) deployed very near the calving front of Bowdoin Glacier in Greenland. The OBS tremor data (~3.5-14 Hz) show remarkable correlation with glacier surface velocities, much higher than correlation with strain rate, suggesting that the tremor is directly related to slip speed rather than strain rate. The findings are very exciting especially in their constraint on subglacial physical processes, and demonstrate the usefulness of observing seismic tremor with near-glacier bed seismic deployments. After very minor revisions, the manuscript will be a very interesting contribution that will be of wide interest to the community. I further note that the analysis is well described, there is sufficient description to reproduce all of their figures, and there is truly no comparable study in the existing literature.

L48 The article “Marine ice sheet profiles and stability under Coulomb basal conditions” by Tsai, Stewart and Thompson (J Glaciol., 2015) would be very appropriate to cite here as a work on the same topic that predates the citations provided.

We have included this overlooked reference, as suggested.

L107-109 Unclear how this statement can be substantiated or what the implication is. Please clarify.

We have placed this sentence before the previous one and added a reference to Fig. 2a to continue the argument that a non-flat spectrum is due to microseism.

Fig3 Why is GPS limited to such a short time? Is the end time when it calved and was lost? I don't question there was a good reason, but it seems strange to not comment on.

We have commented on this logistical/physical constraint in the Introduction.

L146-148 I did not understand the logic of how this statement is related to the previous sentence. Clarifying would be appreciated.

Clarification has been made, with a reference to related to-be-shown analyses being included, and an additional sentence added to aid reader understanding (partly as a response to a similar comment by another Reviewer).

L196 “indicates” is too strong and unsubstantiated. “suggests” would be fine. It is certainly possible that the phenomena are similar, but if the authors want to use “indicates” they need to demonstrate that tremor is generated by friction and high pore pressures, neither of which they appear to have direct evidence for. Perhaps this sounds like nitpicking to the authors, but it is my opinion that their exceptional data need not and should not be overinterpreted. Related to this, the frequency content may be more related to the distance than any similarities in the physics.

Thank you for such precise advice; this word choice was not obvious to us as to non-native-English speakers. We have corrected this and acknowledged the frequency content remark in the main text (in response to a comment by another Reviewer).

L228-232 Again, the 2015 J Glaciol. article cited above is very relevant and predates the citations provided. I encourage the authors to include this paper in their discussion.

This article has been included in the corresponding context.

-Victor Tsai

We thank Dr. Tsai for feedback and help with polishing the paper.

Reviewer #3 (Remarks to the Author):

This manuscript documented possible tremor like signals associated with glacier movements in Greenland as recorded by an OBS. There appears to be a very clear correlations between the tremor signal and ice speed. The authors “proposed that the degree of glacier sliding can be inverted from the observed seismic noise”, and their approach may provide a new way to monitor calving-front processes. Overall I found this manuscript to be written well, and the results are interesting and exciting to justify a publication. However, I felt that the current version did not clearly convince me that the observed signals are similar to (or different from) the non-volcanic tectonic tremor observed at subduction zone and other tectonic settings. I believe that most of them can be accounted by replotting the observations (and rewriting), although the lack of additional OBS recordings is an issue that can only be addressed by future deployments. Below are my specific comments:

We are delighted to hear that the Reviewer finds the results exciting. We hope our replies, revisions, and new plots now present a stronger and easier-to-follow case. We propose a novel method for polar geophysics, and without our pioneering study it is difficult to justify additional deployments. Hopefully, it will engage and bridge at least three otherwise disconnected communities of glaciologists, marine seismologists, and slow-earthquake geoscientists.

1. Due to its non-emergent features, tremor is relatively hard to observe, and typically multiple-station recordings are needed to demonstrate moveout and locate the source, before making further interpretations.

We agree and have demonstrated by our signal processing that it is difficult to extract the tremor from the background seismic noise. In the Conclusions, we mention that, hopefully, our study will motivate further deployments “to reveal tremor location and migration”. Our interpretations are (1) constrained by unique GPS-data representing basal slip (as detailed below), which points to a relatively small section of the glacier being able to serve as a source; and (2), supported by detailed discussion of similarities suggested by most recent laboratory, field, and modeling studies. We note that, even in multi-station tectonic-tremor studies somewhat predating glaciology by ~18 years of progress, the revolutionary interpretation of continuous tremor as a proxy for subduction-zone displacement rate was made without demonstrating move-out or locating the source (*Rouet-Leduc et al., Nature Geoscience, 2019*). We believe that detecting this novel signal is the first milestone.

However, in this study, only one OBS station is used. Hence, there is no way to confirm the observed seismic signals, and locate the seismic source, making the results less convincing.

We agree that location capability is limited. However, we find this criticism of there being “no way to confirm the observed seismic signals”, which were well correlated with *in situ* observed ice speed, to contradict the original contents of the paper. Initially, we showed an independent reproduction of our key finding, by presenting that even in our oldest records (by surface stations in 2015) the tremor could be recognized (when noise conditions were low). Another Reviewer has acknowledged that the latter point is crucial evidence and recommended bringing it forward. We hope the corresponding modifications avoid any misunderstanding.

Since there are several nearby stations on land, perhaps the authors can at least try to plot a zoom-in waveforms (say within 1-2 hrs of a particular day) to demonstrate whether there are (or a lack of) some coherent seismic waves at these stations.

Thank you for suggesting this type of illustration. Such visual inspection of the data was undertaken and an example is provided below. It demonstrates that the noise-floor variation becomes hidden behind local transient events. Therefore, it is difficult to expect and recognize any obvious waveform similarity in the frequency ranges of interest (e.g., 3.5-7.0 Hz). Moreover, it is known that the coherency of seismic waveforms among stations decreases for high frequencies due to scattering by heterogeneities (e.g., Dainty & Toksöz, BSSA, 1990). However, this does not mean that strong local events can't be observed across all stations.

In addition, the authors should also mention about the potential issue with single-station observations in the discussion part of the manuscript.

At the end of the paper, we have mentioned that (1) experiments with several OBS systems would indeed be helpful, and (2) due to the extreme costs of such deployments (e.g., a modern OBS by Güralp/UK costs ~ USD 185,000) single-station experiments might be more feasible.

2. I would like to see a full spectrogram plot (i.e., frequency versus time) for the entire time period of the OBS recording (and perhaps several on-rock stations) to show readers the full information of what were recorded by these seismic stations (rather than only showing the PSD versus time for selected frequency range). A full spectrogram plot would be more informative and hopefully would show either similar or potentially more information than currently shown in Figure 2. This can be easily done using Matlab's specgram function or the continuous wavelet transform (ctw) command.

To provide such basic visualization of the data, full-duration spectrograms were included for all five considered stations as Supplementary Information (Suppl. Fig. 3). In the main text we summarize the discovered signals to maintain a focused message, without overwhelming the reader with broad and intense seismic activity near the calving front, which hides the signal of key interest.

3. Additional information is needed for readers to better understand the experiment setup environment and details of the study region. For example, do we know where the grounding line is, and how far it is from the OBS station? In addition, do we know the rock type and sediment thickness at the ocean bottom?

In this regard, we have provided a new paragraph in the Methods. According to our geodesic observations, Bowdoin Glacier is nearly grounded and has not developed a floating ice tongue. Therefore, the calving front corresponds closely to the grounding line. Similar to many other dangerous-to-approach marine-terminating glaciers, we do not fully understand the underlying local geology. However, we are glad to use this opportunity to highlight that our approach has important implications for addressing this gap. Specifically, we have explained that we included an estimate of the sediment thickness based on H/V spectral ratio analysis.

4. The authors mentioned in the abstract that "the tremor was also confirmed via an analysis of the seismic waveforms from surface stations". If I remembered correctly, the authors stated that the tremors were hard to observe from surface stations, and did not show any correlations with the GPS movements. Please clarify.

Our initial text and plots showed good (e.g., $r = 0.77$) correlations between the surface seismic stations and GPS data and discussed in detail the reasons for cases of a poor correlation (e.g., $r = 0.2$; due to wind noise). We imply this in the Abstract.

If the tremors can be observed by both the surface stations and OBS, then some plots/analysis as suggested in comment 1 should be performed.

Please find such plots and our response to this with Comment 1. Tremor presence across the stations is seen as noise-floor variation, hidden below transient events local to each station.

Finally, the authors stated that “The signal resembles the nonvolcanic tremor ...”. I wonder if they can make some plots to show some representative time series recorded by this station and those observed at subduction zones.

Our interpretation stems from a detailed discussion of features, similar physical processes, and experimental evidence rather than from the apparent similarity of time-series alone. If there were previous tectonic time-series that could be placed next to the glacier series, we agree that this would be a useful addition. However, for this study, we find this idea difficult to accommodate, for the following reasons:

- First, the continuous glacial tremor is hidden in transient events and therefore requires the signal processing described in the paper. Similarly to this, the closest tectonic example, the continuous chatter of the Cascadia subduction zone, was revealed by substantial signal de-noising based on machine learning (*Rouet-Leduc et al., 2019*).
- Second, reproduction of such subduction-zone signals within this paper requires us to either (a) reproduce previously published materials, or (b) initiate and include some different subduction zone projects conducted with other collaborators. We do not find the latter option as feasible. Furthermore, perhaps, the journal might consider permitting us the former option, to reproduce previously published plots (e.g., see Fig. 2 below from *Rouet-Leduc et al., 2019*). However, in our opinion, that would reduce the authenticity of our original contribution, and is uncommon for the journal.

Fig. 2 | Estimating the GPS displacement rate from the continuous seismic data. **a**, Smoothed GPS displacement rate (red). Estimate (est.) from the machine-learning model (blue) with estimation intervals (shades of blue) using statistical features of the full continuous seismic data (Methods) from the six considered seismic stations (PGC, NLLB, SNB, VGZ, PFB and MGB) as input. The figure shows the testing set, for which the algorithm only has access to the seismic data (for example, as in **b**). Gaps indicate missing data. **b**, Continuous seismic data from station NLLB over the same time interval. **c**, Distribution of observed versus predicted displacement rates; the contours show empirical isodensity, from 10 to 90%. The Pearson correlation coefficient between the estimates and actual displacement rate is 0.66, which shows that the continuous seismic waves contain rich information about the fault's state, apparently at all times. **d**, Schematic of the approach.

In addition, in source seismology communities, we typically use “tectonic tremor”, rather than “nonvolcanic tremor” these days (<https://pubs.er.usgs.gov/publication/70192475>).

Thank you for explaining this and the reference. Initially, for consistency with cited literature, we adopted “non-volcanic tremor” from the original source (Obara, 2002). In the revision, we have introduced the suggested synonym, cited the provided reference (i.e., Shelly, 2016), and used “tectonic tremor” in the abstract.

5. In the end of the abstract, the authors stated that “the degree of glacial sliding can be inverted from the observed seismic noise”. I am not sure what they meant by “the degree of glacial sliding”. Indeed, we saw a positive correlation between the surface GPS recording and the tremor-like signals. But surface GPS movement does not exactly correspond to the degree of glacial sliding, right? Some clarification is needed.

We have improved the language (i.e., “the sliding velocity”) and, as recommended, have clarified (in Methods) that GPS recordings correspond very closely to basal sliding, because near the calving front there is very little internal deformation at play, as commonly assumed in models (Tsai et al., 2015, Stearns and van der Veen, 2018) and known from our direct borehole deformation measurements at Bowdoin Glacier (Seguinot et al., 2018, 2020).

6. The authors used the term “signal” and “noise” interchangeably in the abstract and main text. It is clear that noise can be turned into signals. But I think that some simple definition at the beginning of the main text could be useful and will not confuse the readers.

We have acknowledged this issue in the Introduction and provided a reference to a recent book discussing this terminological dispute (Seismic Ambient Noise, 2019).

7. Line 217, the authors wrote “Glacier tremor is stronger and more clearly correlated with the seismic data than subduction zone tremor [4]”. I am not sure what did they refer to as “seismic data”. Perhaps rather than “seismic data”, the authors meant “GPS data”?

Yes, it should be “GPS data”; thank you for noticing this.

8. The following references on the observations of seismic tremor associated with glacial movements, and reviews of global observations of fast and slow earthquakes are worth citing: Winberry J. P., S. Anandakrishnan, D. A. Wiens, and R. B. Alley (2013), Nucleation and seismic tremor associated with the glacial earthquakes of Whillans Ice Stream, Antarctica, *Geophys. Res. Lett.*, 40, 312–315, doi:10.1002/grl.50130.
Peng, Z. and J. Gomborg (2010), An integrative perspective of coupled seismic and aseismic slow slip phenomena, *Nature Geosci.*, 3, 599–607, doi: 10.1038/ngeo940.

As suggested, these references have been added.

We thank Reviewer #3 for helping us to improve the quality of the paper.

REVIEWER COMMENTS

Reviewer #1 (Remarks to the Author):

The authors have done an excellent job improving this manuscript. They have added previously supplementary seismic data to the main text; this data supports their arguments and is a good addition. They have also more clearly outlined why they believe the seismic signal they are interested in is not of hydrologic origin. These changes successfully address most of the main points in my review.

I have a few additional suggestions to further clarify this manuscript and make it easier to understand.

Regarding figures:

1. I would suggest adding the tidal data to the figures in the main text. I found myself wanting to compare the signals to the tides every time I looked at them, but it's hard because the tidal information is both plotted at a different scale and hidden in the supplement.

2. I like the addition of more seismic data to the main text, though it is a lot of figures and not all parts of them support your conclusions. Instead of just including all of the on-ice and on-rock data in the main text as separate figures, I would suggest something similar to the following: (a) Show in a separate figure just the on-rock data that best supports your conclusion that the tremor signal relates to basal slip. (b) Create a summary figure that plots (for each band in a different panel, and for each year of data, as makes sense) just the running noise floor for all seismic sites on the same axis, and the GPS and tidal data you are comparing to. You might also find a way to include relevant cross plots of Ice Speed vs. Normalized min PSD (ex. Fig 3b) to show which data streams are correlated. Your current Supplementary Figure 5 is similar to what I'm thinking.

The point would be to make it easier to compare your data streams without flipping between figures. I realize not all the data streams overlap, and this might not make sense, so it is up to the authors.

3. Related: In general, a lot of the figures show data that distract from the main conclusion that a specific tremor band is related to basal slip (for example, normalized PSD for data from 0.1-0.6 Hz). I suggest the authors be thoughtful about which data to keep in the main text, and which to keep in the supplement.

I find the section comparing the glacier tremor to slow earthquake processes still somewhat hard to follow logically, especially the last paragraph.

Throughout - add years to dates.

Other line comments: (Line numbers refer to the version with tracked changes included)

34: should this be "the connection between slow and fast earthquakes"? Slow earthquakes can also be large

170: "processes" is vague; suggest clarifying again the specific process you think drives the seismicity, and

that in this paragraph you're describing why the signal of interest can't be from surface crevassing

187: I don't really see the double-hump feature. Maybe you could annotate the figure to point it out?

215-218: Did this calving event make a glacial earthquake detectable in the far-field? If not, then the calving event was probably not very energetic (relative to how energetic calving events CAN be) and this claim is an overstatement.

232-235: Again, I don't really see morning peaks?

293: "...off-ice stations do not miss high-frequency processes..." I don't understand what this means?

321-323: I don't understand how the contrast in timescales helps explain why tectonic GPS are less-correlated with tremor.

336-347: I don't quite follow the logic here. Some of the arguments seem to undercut your suggestion that seismic radiation could scale with sliding speed?

Reviewer #3 (Remarks to the Author):

The revised version addressed most of the previously raised comments. I only have a few very minor comments/suggestions that the authors can decide whether they need to be addressed later. I recommend accepting it as is (or after a minor revision).

1. Figures 4-7: please move panels b and c up and closer to panel a so that the figures look tighter. In addition, since they show similar information, I wonder if it would make sense to keep on in the main text, and the rest to be in the method section?
2. Page 12 of the rebuttal letter (and supplementary Figure 1): perhaps I am beating a dead horse here, but I wonder if the authors can add a smoothed envelope function for each band-pass-filtered seismic traces, and plotted them together to see if there might be any coherent phases (e.g., similar to supp. Figure 1 of Obara (Science, 2002) paper). In addition, it would be useful to add detailed information on the corresponding day in July in the figure caption.
3. Page 11 in the rebuttal letter: just a response to the author's statement in Rouet-Leduc et al., 2019, they did not demonstrate the move-out or locate the tremor sources to make their main conclusion (continuous tremor can be used as a proxy for subduction-zone displacement rate). I agree with this statement by itself. However, tremor has been well studied and located by other earlier studies. Hence, there is no need for them to perform these tasks. However, because this phenomenon is relatively new in the field of glaciology, it is definitely worth to demonstrate that these tremor sources indeed can be recorded by multiple stations and do show consistent movements. Perhaps the authors can try with the envelope functions as described above to see if any coherence noise-like signals can be observed?

Re-submitted: 5 Feb 2021, Invited-to-revise: 7 Apr 2021

We thank two anonymous reviewers for evaluating the revised version of our manuscript and providing additional suggestions. We are delighted to hear it successfully addresses most of their comments and that they find it has improved. The updated version of our manuscript is enclosed, and our point-by-point responses are given below (shown in bold; revisions are highlighted in the manuscript).

Yours sincerely,
Evgeny Podolskiy and co-authors

REVIEWER COMMENTS:

Reviewer #1 (Remarks to the Author):

The authors have done an excellent job improving this manuscript. They have added previously supplementary seismic data to the main text; this data supports their arguments and is a good addition. They have also more clearly outlined why they believe the seismic signal they are interested in is not of hydrologic origin. These changes successfully address most of the main points in my review.

I have a few additional suggestions to further clarify this manuscript and make it easier to understand.

Thank you!

Regarding figures:

1. I would suggest adding the tidal data to the figures in the main text. I found myself wanting to compare the signals to the tides every time I looked at them, but it's hard because the tidal information is both plotted at a different scale and hidden in the supplement.

Different glaciers respond differently to tidal forcing, and a lack of *in situ* records of glacier movement is a weakness that has led to multiple hypothesis regarding the interpretation of tide-modulated seismicity. As explained in the paper, we analyzed Bowdoin tides in our previous studies, so we emphasized GPS data here and have provided tide data (downloaded from the Internet) in the Supplement. Figures that compare noise with other time-series contain up to four to seven overlapping curves. This, along with the addition of tidal data, with the extra y-label and y-axis labels, makes such figures confusing, which we prefer to avoid. As a compromise, we have moved the tidal-data figure from the Supplement to the main text.

2. I like the addition of more seismic data to the main text, though it is a lot of figures and not all parts of them support your conclusions. Instead of just including all of the on-ice and on-rock data in the main text as separate figures, I would suggest something similar to the following: (a) Show in a separate figure just the on-rock data that best supports your conclusion that the tremor signal relates to basal slip. (b) Create a summary figure that plots (for each band in a different panel, and for each year of data, as makes sense) just the running noise floor for all seismic sites on the same axis, and the GPS and tidal data you are comparing to. You might also find a way to include relevant cross plots of Ice Speed vs. Normalized min PSD (ex. Fig 3b) to show which data streams

are correlated. Your current Supplementary Figure 5 is similar to what I'm thinking. The point would be to make it easier to compare your data streams without flipping between figures. I realize not all the data streams overlap, and this might not make sense, so it is up to the authors.

Thank you for such detailed technical advice and for leaving the potential changes to our discretion. The discussed figures were added following a specific request from the first review to support our main interpretation and to illustrate the performance of stations operating in different years and under different conditions. As we partition the noise into different bands separately for each station and consider different GPS stations, it is appropriate to present them separately. This also makes it easier to organize figure captions that mention particular events and instrument IDs; as we now describe the figures in order in the text, we find this approach logical. Moreover, the flow of the main text, which has now been approved following comments by the three referees, follows the illustration order and we have no intention of setting on-rock data aside, as the on-ice data also contain the signal but it is blurred by other sources, as explained. As the Reviewer noted, some data streams do not overlap (or belong to different years, 2015 or 2019) and experiments "on the same axis" are difficult to argue for. This might be appropriate in applying an overview (as presented in Supplementary Fig. 2), but it would make fine details unrecognizable and corresponding discussion in the main text difficult to follow.

3. Related: In general, a lot of the figures show data that distract from the main conclusion that a specific tremor band is related to basal slip (for example, normalized PSD for data from 0.1-0.6 Hz). I suggest the authors be thoughtful about which data to keep in the main text, and which to keep in the supplement.

The number of subplots was reduced, by moving 8 of them in the Supplement.

I find the section comparing the glacier tremor to slow earthquake processes still somewhat hard to follow logically, especially the last paragraph.

Related line comments have been addressed. The last paragraph, which included text suggested by another reviewer, has been re-written.

Throughout - add years to dates.

We have added the year information to figure captions as necessary.

Other line comments: (Line numbers refer to the version with tracked changes included)

34: should this be "the connection between slow and fast earthquakes"? Slow earthquakes can also be large

We agree, and have corrected this.

170: "processes" is vague; suggest clarifying again the specific process you think drives the seismicity, and that in this paragraph you're describing why the signal of interest can't be from surface crevassing

We have already stated in the previous paragraph in the manuscript what would be described in this one. Therefore, for conciseness and clarity, we changed "the

processes responsible for this temporal change of ambient noise levels” to “the observed continuous seismic signal”.

187: I don't really see the double-hump feature. Maybe you could annotate the figure to point it out?

As suggested, we have annotated Supplementary Fig. 4.

215-218: Did this calving event make a glacial earthquake detectable in the far-field? If not, then the calving event was probably not very energetic (relative to how energetic calving events CAN be) and this claim is an overstatement.

Yes, this calving event generated a glacial earthquake in the far field, as mentioned in the text and highlighted in Supplementary Fig. 5.

232-235: Again, I don't really see morning peaks?

Following the above advice, the peaks were also labeled in the corresponding Fig. 5a.

293: "...off-ice stations do not miss high-frequency processes..." I don't understand what this means?

In the context of attenuation, we have re-written this as “high-frequency seismicity”, and also referred to Supplementary Fig. 4.

321-323: I don't understand how the contrast in timescales helps explain why tectonic GPS are less-correlated with tremor.

We have stated there may be a reduction in correlation due to noisy episodes, which are likely to occur in the longer term.

336-347: I don't quite follow the logic here. Some of the arguments seem to undercut your suggestion that seismic radiation could scale with sliding speed?

This paragraph refers to the recent debate in glaciology regarding the basal sliding of rapid ice flows, and we mentioned that our interpretations may appear inconsistent with some of the literature. However, more recent studies indicate that this is not the case (as detailed in the text). We hope our revised and edited manuscript is now easier to follow.

We thank again Reviewer #1 for the feedback and help in improving the paper.

Reviewer #3 (Remarks to the Author):

The revised version addressed most of the previously raised comments. I only have a few very minor comments/suggestions that the authors can decide whether they need to be addressed later. I recommend accepting it as is (or after a minor revision).

Thank you!

1. Figures 4-7: please move panels b and c up and closer to panel a so that the figures look tighter. In addition, since they show similar information, I wonder if it would make sense to keep on in the main text, and the rest to be in the method section?

As suggested, we have moved the panels closer together. Moreover, the number of subplots was reduced, by moving 8 of them in the Supplement.

2. Page 12 of the rebuttal letter (and supplementary Figure 1): perhaps I am beating a dead horse here, but I wonder if the authors can add a smoothed envelope function for each band-pass-filtered seismic traces, and plotted them together to see if there might be any coherent phases (e.g., similar to supp. Figure 1 of Obara (Science, 2002) paper). In addition, it would be useful to add detailed information on the corresponding day in July in the figure caption.

Thank you for providing extra thoughts in this regard. The Introduction highlights that we are dealing with an extremely noisy near-field environment, which prevented us from identifying the tremor directly. The tremor-like signal is so weak compared with the transient background noise field that it had to be extracted from the long-term noise floor variation. This involves relatively long time-windows for PSD computation (30 s; already longer than any time lag between our proximally located stations), and smoothing over 30-minutes-long sliding windows. Such long time-scales and the overwhelming number of local transient events mean that if we used envelopes we would no longer observe the process of interest. Coherent short transients (e.g., due to calving events) might be expected for which envelopes would be helpful, but this is not the aim of the study.

As suggested, the date is provided in the figure caption.

3. Page 11 in the rebuttal letter: just a response to the author's statement in Rouet-Leduc et al., 2019, they did not demonstrate the move-out or locate the tremor sources to make their main conclusion (continuous tremor can be used as a proxy for subduction-zone displacement rate). I agree with this statement by itself. However, tremor has been well studied and located by other earlier studies. Hence, there is no need for them to perform these tasks. However, because this phenomenon is relatively new in the field of glaciology, it is definitely worth to demonstrate that these tremor sources indeed can be recorded by multiple stations and do show consistent movements. Perhaps the authors can try with the envelope functions as described above to see if any coherence noise-like signals can be observed?

Thank you for reading our replies and explaining this point. The paper demonstrates that temporal changes in ambient noise level can be recorded by different stations in different years. The precise location of the source of this long-term, semi-diurnal tremor is interesting. However, as explained in our response to the previous comment, there is a mismatch between time-scales in our study and envelope functions would shift attention to other transient phenomena. We hope this study will justify detailed follow-up in, for example, comparing time-lapse imagery and other data with located transient events for classifying their types and relevance to the long-term tremor using larger networks.

We thank Reviewer #3 for their ideas and help in improving the paper.

REVIEWERS' COMMENTS

Reviewer #3 (Remarks to the Author):

All my comments have been properly addressed. I recommend accepting the manuscript as is.

Re-submitted (2nd time): 14 April 2021, Invited-to-revise: 18 May 2021

REVIEWER COMMENTS:

Reviewer #3 (Remarks to the Author):

All my comments have been properly addressed. I recommend accepting the manuscript as is.

Thank you!

**Yours sincerely,
Evgeny Podolskiy and co-authors**